# Comprehensive profiling of migratory primordial germ cells reveals niche-specific differences in non-canonical Wnt and Nodal-Lefty signaling in anterior vs posterior migrants

**Rebecca Garrett Jaszczak[†], Jay W Zussman[†], Daniel E Wagner, Diana J Laird***

Eli and Edythe Broad Center for Regeneration Medicine and Stem Cell Research and Department of Obstetrics, Gynecology and Reproductive Science, San Francisco, United States

## eLife Assessment

This revised study provides **fundamental** insights into the differences in migratory primordial germ cells based on their anterior or posterior location. Through **convincing** methodology and analysis of single-cell RNA sequencing of an exceptionally large number of migratory primordial germ cells and surrounding somatic cells, the novel findings and datasets generated from this study provide many hypotheses of interest to germ cell biologists.

**\*For correspondence:**
Diana.laird@ucsf.edu

[†]These authors contributed equally to this work

**Abstract** Mammalian primordial germ cells (PGCs) migrate asynchronously through the embryonic hindgut and dorsal mesentery to reach the gonads. We previously found that interaction with different somatic niches regulates mouse PGC proliferation along the migration route. To characterize transcriptional heterogeneity of migrating PGCs and their niches, we performed single-cell RNA sequencing of 13,262 mouse PGCs and 7868 surrounding somatic cells during migration (E9.5, E10.5, E11.5) and in anterior vs posterior locations to enrich for leading and lagging migrants. Analysis of PGCs by position revealed dynamic gene expression changes between faster or earlier migrants in the anterior and slower or later migrants in the posterior at E9.5; these differences include migration-associated actin polymerization machinery and epigenetic reprogramming-associated genes. We furthermore identified changes in signaling with various somatic niches, notably strengthened interactions with hindgut epithelium via non-canonical WNT (ncWNT) in posterior PGCs compared to anterior. Reanalysis of a previously published dataset suggests that ncWNT signaling from the hindgut epithelium to early migratory PGCs is conserved in humans. Trajectory inference methods identified putative differentiation trajectories linking cell states across timepoints and from posterior to anterior in our mouse dataset. At E9.5, we mainly observed differences in cell adhesion and actin cytoskeletal dynamics between E9.5 posterior and anterior migrants. At E10.5, we observed divergent gene expression patterns between putative differentiation trajectories from posterior to anterior, including Nodal signaling response genes *Lefty1, Lefty2,* and *Pycr2* and reprogramming factors *Dnmt1, Prc1,* and *Tet1*. At E10.5, we experimentally validated anterior migrant-specific *Lefty1/2* upregulation via whole-mount immunofluorescence staining for LEFTY1/2 and phosphorylated SMAD2/3, suggesting that elevated autocrine Nodal signaling in migrating PGCs occurs as they near the gonadal ridges. Together, this positional and temporal atlas of mouse PGCs supports the idea that niche interactions along

the migratory route elicit changes in proliferation, actin dynamics, pluripotency, and epigenetic reprogramming.

## Introduction

The mammalian germline is initially specified as a small pool of primordial germ cells (PGCs) (*Lawson and Hage, 1994*), which will give rise to oocytes or spermatozoa and transmit genetic information across generations. A founding population of ~40 mouse PGCs is specified at E7.25 (*Anderson et al., 2000*, *Canovas et al., 2017*, *Ginsburg et al., 1990*) from the epiblast driven by bone morphogenetic protein (BMP) signaling from adjacent extraembryonic tissues (*Lawson et al., 1999*; *Ying et al., 2000*). Shortly after specification, PGCs begin migrating through the hindgut, moving anteriorly toward the forming gonadal ridges. Signaling cues such as Cxcr4 and KitL are known PGC chemoattractants and contribute to this journey while also promoting PGC survival (*Pesce et al., 1993*; *Farini et al., 2007*). Studies observing this migratory process have characterized distinct behavioral patterns in the migratory population. Time-lapse imaging across the period of mouse germ cell migration from E9.5 to E11.5 showed that PGCs early in their migration are motile within the hindgut, moving both with the morphogenetic changes of the developing embryo and anteriorly in their own right (*Molyneaux et al., 2001*). At E9.5, a behavioral change occurs, as some PGCs begin directed migration out of the hindgut and into the dorsal mesentery tissues. This targeted migration continues through E10.5, by which point the leading PGC migrants home in on the developing gonadal ridges (*Molyneaux et al., 2001*). At E10.5, PGCs are located within diverse tissue niches, with migratory leaders arriving in the gonadal ridges, actively migrating cells still in the dorsal mesentery, and migratory laggards remaining in the hindgut. By E11.5, most PGCs have reached the gonadal ridges; however, a population of PGCs remains medial to the gonad and is considered ectopic (*Runyan et al., 2008*).

Though the spatiotemporal distribution of migrating PGCs is well characterized, molecular features distinguishing leading vs lagging migrants remain poorly understood. Work from our lab demonstrated that early in migration, PGCs receive non-canonical WNT (ncWNT) signaling which suppresses their proliferation within the hindgut. As they leave the hindgut and progress into the mesentery and gonad, they display progressively greater canonical WNT signaling and proliferate more rapidly, reaching maximal proliferation within the gonad (*Cantú et al., 2016*). Epigenetic reprogramming occurs concurrently with PGC migration from E9.5 to E11.5, during which global DNA demethylation erases genomic imprints and enables later upregulation of gametogenesis and meiosis-specific genes (*Hill et al., 2018*; *Hargan-Calvopina et al., 2016*; *Seisenberger et al., 2012*). It remains unknown how shifting niche interactions throughout migration may alter PGC transcriptomes and epigenetic reprogramming, or how these influences might differentially affect earlier vs later migrants.

Investigating transcriptional differences within the PGC population during migration has been challenging as the cell population at migratory timepoints starts out quite small with just ~200 cells present at E9.5 (*Durcova-Hills et al., 2003*), ~1000 cells present at E10.5 (*Saiti and Lacham-Kaplan, 2007*) and ~2600 cells present at E11.5 (*Laird et al., 2011*). Other groups have harnessed single-cell RNA sequencing to investigate germ cell development (*Stévant et al., 2019*; *Hermann et al., 2018*; *Wang et al., 2020*; *Li et al., 2017*; *Mayère et al., 2019*; *Zhao et al., 2020*; *Alexander et al., 2024*; *Garcia-Alonso et al., 2022*); however, these studies have precluded in-depth study of migratory PGC behavior and transcriptional dynamics within single timepoints due to the relatively small numbers of migratory PGCs in their respective datasets.

By pooling many somite-matched embryos, we successfully profiled large populations of PGCs from E9.5, E10.5, and E11.5 embryos, separating anterior and posterior embryo halves in order to isolate leading from lagging migrants and facilitate direct transcriptional comparisons between these populations. We also profiled representative cells from PGC niches along the migratory path. By investigating cell-cell communication pathways between migrating PGCs and their somatic niches, we uncover interactions that influence PGC differentiation over the course of migration.

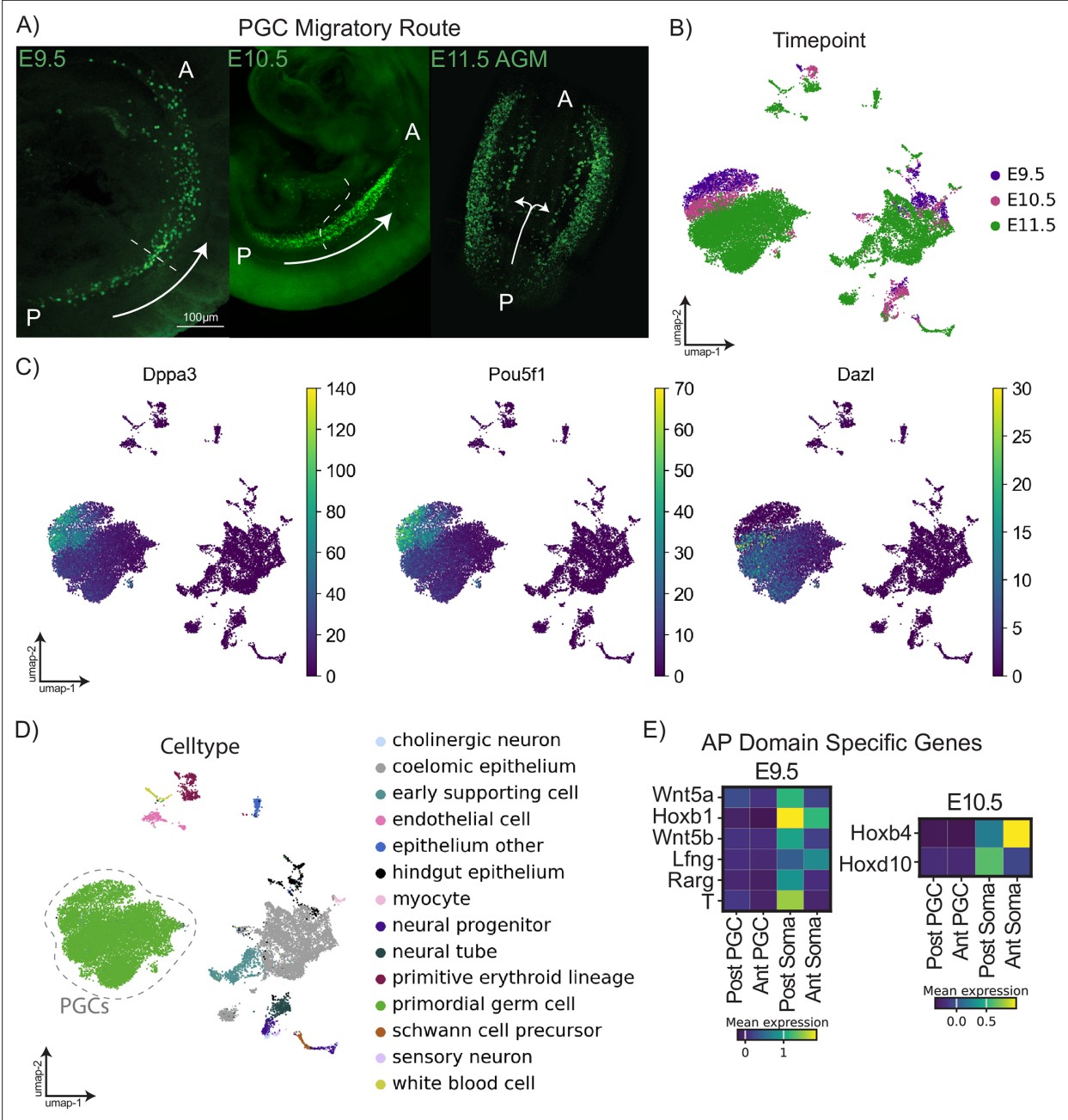

**Figure 1.** Developmental and transcriptional profile of E9.5-E11.5 migratory mouse primordial germ cells. (**A**) Confocal microscopy images of whole-mount mouse embryos at E9.5, E10.5, and E11.5 (left to right). Primordial germ cells (PGCs) are visualized via the Pou5f1-ΔPE-eGFP transgene (MGI:3057158) stained with anti-GFP antibody (ab13970). P indicates posterior of migratory stream; A indicates anterior. White arrows indicate the general direction of PGC migration by timepoint. (**B**) Uniform Manifold Approximation and Projection (UMAP) plot of all cells profiled in this dataset (from pooled pre-sex specification XY and XX embryos) colored by timepoint arranged using STITCH by timepoint (see Materials and methods - Batch correction). (**C**) UMAP plots of all cells colored by log-normalized expression of marker genes relevant in PGC development. (**D**) UMAP plot of all cells profiled in this dataset colored by annotated cell type. (E) Matrixplot of genes with anteroposterior domain-specific expression at E9.5 and E10.5.

## Results

### A comprehensive positional survey of migrating PGCs and their niches

To ascertain the transcriptional dynamics within mouse PGCs across the migratory time period, we performed droplet-based single-cell RNA sequencing to create libraries from E9.5, E10.5, and E11.5 mouse embryos, to span the onset of directed migration (*Molyneaux et al., 2001*) and homing of

PGCs into the gonadal ridges (*Figure 1A*). Since PGCs are few in number at early migratory time-points, we used the Oct4-ΔPE-eGFP reporter (MGI:3057158; *Szabó et al., 2002*) to purify PGCs from multiple age-matched embryos via FACS (see Materials and methods). We constructed sequencing libraries from bisected E9.5 and E10.5 embryos (see Materials and methods) to facilitate comparisons between the transcriptomes of leading (anterior) and lagging (posterior) migrants (*Figure 1A*, first two panels). At E11.5, 90–95% of PGCs have arrived at the forming gonadal ridge (*Laird et al., 2011*) with some remaining in the midline between the gonadal ridges (*Figure 1A*, third panel). We purified GFP+ PGCs from the aorta-gonad-mesonephros (AGMs) to enrich both for successful migrants and any PGCs remaining in the midline and mesentery proximal to the gonads. To survey the supporting somatic cell niches at each migratory timepoint and anatomical position, we included GFP-negative somatic cells from the same sort.

In total, we profiled 21,205 cells (13,262 PGCs and 7943 somatic cells) from all migratory time-points (*Figure 1B*). Results were visualized by Uniform Manifold Approximation and Projection (UMAP) embeddings. We saw clear expression of early PGC markers *Pou5f1* and *Dppa3 (Stella)* among E9.5 and E10.5 PGCs and increasing expression of *Dazl* with PGC developmental stage (*Figure 1C*). Using automated cell-type identification from previously published datasets (see Materials and methods), we identified the following somatic populations: coelomic epithelium, bipotential early gonadal supporting cells, neural tube, hindgut epithelium, primitive erythroid lineage, endothelial cells, neural progenitors, other epithelia, Schwann cell precursors, white blood cells, cholinergic neurons, myocytes, and sensory neurons (*Figure 1D*).

To validate the positional separation of the cells included in the anterior vs posterior libraries, we examined genes with established roles in caudal patterning of the embryo. As expected, *Wnt5a*, *Wnt5b, Hoxb1, Rarg, Lfng*, and *T* (*Gofflot et al., 1997*) were highly enriched in somatic cells of our posterior E9.5 sample (*Figure 1E*). At E10.5, Hoxd10 expression is known to span somites 31–41 (*Burke et al., 1995*; *Wellik, 2007*) and was higher in somatic cells of our posterior libraries, which contained tissue from somites greater than 21. *Hoxb4* expression has been previously described from somites 6 to 41 (*Burke et al., 1995*) and was present in both anterior and posterior somatic libraries at E10.5, while expression of *Hoxd10* was mostly restricted to the posterior sample (*Figure 1E*). This analysis supports that anterior and posterior tissues were successfully segregated using our embryo splitting strategy.

## Transcriptomic shifts over developmental time in migratory and post-migratory PGCs

All together, we examined 1268 PGCs at E9.5, 1664 PGCs at E10.5, and 10,330 PGCs at E11.5. This large number of PGCs permitted assessment of migratory PGC heterogeneity across developmental time. After pairwise differential expression testing among PGCs at each developmental timepoint, we conducted Gene Ontology (GO) or Gene Set Enrichment Analysis (GSEA) on the resultant lists of differentially expressed genes. E9.5 PGC migrants exhibited an elevated signature of regulation of canonical WNT signaling relative to E10.5 (*Figure 2—figure supplement 1A*). This ontology term result does not indicate the direction of such regulation, but may be consistent with prior findings that WNT signaling regulates proliferative capacity of migrating PGCs (*Cantú et al., 2016*), with increased canonical WNT signaling promoting PGC proliferation later in migration. E9.5 migrants also differentially expressed genes associated with focal adhesion compared to E10.5 (*Figure 2—figure supplement 1B*), suggesting that a shift in cytoskeletal dynamics accompanies the behavior change from chiefly migratory to more proliferative over the course of migration. During PGC migration, progressive genome-wide DNA demethylation permits expression and potential transposition of transposable elements (*Brennecke et al., 2007*; *Teixeira et al., 2017*). Consistent with this, we found piRNA processing was an upregulated GO term at E10.5 compared to E9.5 (*Figure 2—figure supplement 1A*), suggesting that expression of genes relevant for piRNA processing increases during PGC migration despite the fact that mature piRNAs are not detected until E13.5 (*Ernst et al., 2017*; *Ramakrishna et al., 2022*). Additionally, compared to migrating PGCs at E9.5, post-migratory E11.5 germ cells express more genes associated with GO/GSEA terms related to chromatin remodeling (*Figure 2—figure supplement 1C*), consistent with the global demethylation present by E11.5 (*Hill et al., 2018*) and oxidative phosphorylation (*Figure 2—figure supplement 1D*), which may reflect the increased proliferation previously observed within the gonadal niche (*Cantú et al., 2016*).

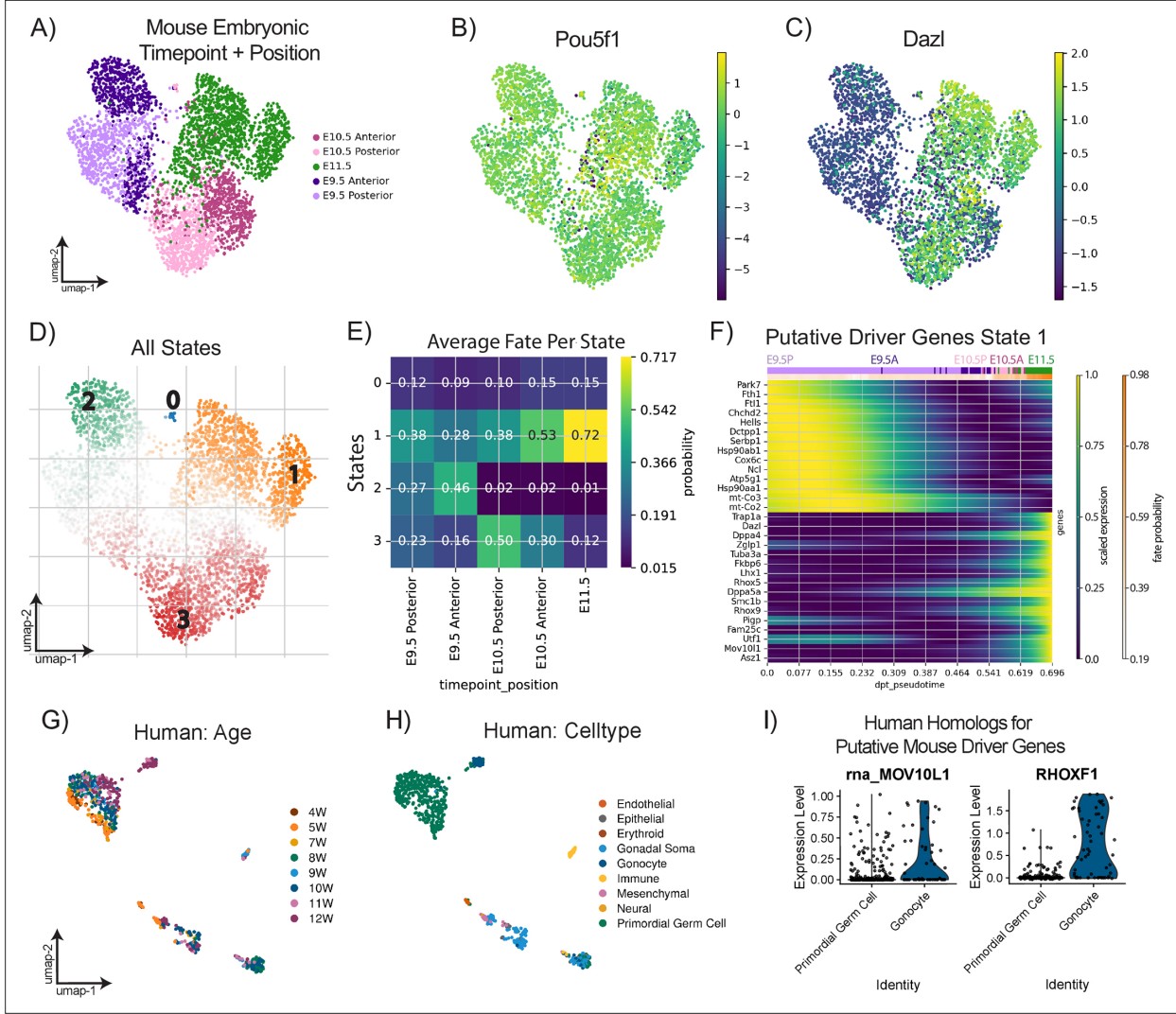

**Figure 2.** Trajectory inference defines terminal states for migratory mouse germ cells and conserved expression of putative driver genes across mouse and human. (**A**) Uniform Manifold Approximation and Projection (UMAP) plot of primordial germ cells (PGCs) downsampled to 1268 cells/timepoint colored by position-time of sample. (**B**) UMAP plot colored by *Pou5f1* gene expression. (**C**) UMAP plot colored by *Dazl* gene expression. (**D**) All initial and terminal states predicted from CellRank transition matrix analysis based on diffusion pseudotime plotted on UMAP representation of downsampled PGCs across timepoints. (**E**) Matrixplot of fate probability of reaching identified terminal states by position-time identity of PGCs. (**F**) Putative driver genes with expression dynamics correlated with terminal state 1. Color bar at top of plot indicates position-time of cells most highly expressing driver genes below; see color key in A. (**G**) UMAP plot of human PGCs and somatic cells from *Li et al., 2017*, colored by gestational week. (**H**) UMAP plot of human PGCs and somatic cells colored by annotated cell types. (**I**) Violin plots of log-normalized expression of selected genes with conserved expression patterns between mouse and human.

The online version of this article includes the following figure supplement(s) for figure 2:

**Figure supplement 1.** Gene ontology and random walk simulations supporting temporal characterization of mouse primordial germ cells with human gene expression by cell type.

Next, we leveraged trajectory inference techniques in CellRank after subsetting the PGC population to include 1268 cells from each timepoint (*Figure 2A*); this subsampling avoided potential biases in embedding graph connectedness resulting solely from differences in PGC numbers from each time-point, and we further regressed out cell cycle phase based on S and G2M scores. As expected for this developmental window (*Saiti and Lacham-Kaplan, 2007*), we observed robust expression of *Pou5f1* across E9.5–11.5 PGCs (*Figure 2B*) and successive upregulation of *Dazl* with increasing devel-opmental time (*Figure 2C*). To better understand likely differentiation trajectories across these key migratory and early post-migratory timepoints, we employed CellRank's pseudotime kernel to build

a transition matrix among PGCs. Simulating random walks of 100 randomly selected E9.5 germ cells for 100 steps revealed that E9.5 posterior PGCs primarily reached endpoints among E11.5 PGCs, with some terminating among E10.5 and others remaining within a terminal state mostly occupied by E9.5 anterior PGCs (*Figure 2—figure supplement 1E*). When this was repeated using CellRank's real-time kernel only, which used only timepoint information and transcriptional relationships to build its transition matrix, all E9.5 PGCs reached final positions among E11.5 PGCs (*Figure 2—figure supplement 1F*). This discrepancy may reflect differences in differentiation potential of some E9.5 PGCs that end in a terminal state among anterior E9.5 PGCs, but could also result from technical batch effects generated during library preparation. These possible interpretations are further discussed in the Discussion section.

We detected initial and terminal states found in E9.5–11.5 PGC data based on diffusion pseudotime with a root cell within the posterior E9.5 sample and CellRank's pseudotime kernel. The predicted initial state was found among E9.5 PGCs (*Figure 2D*, green highlight 2) and was also identified as a putative terminal state. We also found one terminal state at E10.5 (red, 3) and one terminal state at E11.5 (orange, 1). Next, we predicted fate probabilities for each cell not assigned to a terminal state to reach each predicted terminal state (*Figure 2E*). Genes with expression patterns found to be highly correlated with specific terminal states and significantly up- or downregulated during a specific transition from a predicted initial state to a terminal state were considered candidate driver genes of that terminal state.

Since terminal state 1 was found within the E11.5 PGCs in the dataset, we hypothesized that cells with high fate probabilities (*Figure 2E*) for terminal states 1 could represent successful migrants, reaching the gonadal ridge and continuing germline development. This is supported by our discovery of various canonical germ cell development genes as genetic drivers for terminal state 1, including *Dazl, Dppa4, Dppa5a, Lhx1,* and *Rhox* genes (*Lin and Page, 2005*; *Madan et al., 2009*; *Tanaka et al., 2010*; *Tan et al., 2021*; *Figure 2F*). We also uncovered driver genes for state 1 relevant for critical germ cell functions at these timepoints, including *Smc1b*, which is directly involved with epigenetic reprogramming (*Rengaraj et al., 2022*; *Zhao et al., 2017*; *Kim et al., 2011*) and *Mov10l1* and *Asz1*, which were previously implicated in genome defense against retrotransposons during reprogramming (*Rolland et al., 2011*; *Ikeda et al., 2022*; *Ollinger et al., 2008*; *Reichmann et al., 2020*; *Vourekas et al., 2015*). In this analysis, even the developmentally earliest PGCs in the dataset (E9.5 posterior) were assigned high fate probabilities of reaching state 1 (*Figure 2E*); this supports the possibility of an in vivo differentiation pathway linking initial state 3 with terminal state 1 during PGC migration.

To test our findings for conservation between mouse and human embryos, we took advantage of a publicly available dataset containing human (h) PGCs from 4 to 12 weeks (W), which corresponds developmentally to mouse PGCs from E9.5 to E11.5 (*Li et al., 2017*; *Figure 2G*). Like in their mouse counterparts, *POU5F1* marked hPGCs, and a gonocyte cluster was marked by *DAZL* (*Figure 2—figure supplement 1G*). We identified somatic populations similar to those found in our mouse dataset through automated annotation (*Figure 2H*). Next, we explored the expression of genes identified as drivers for terminal state 1 in mouse PGCs. We found that *RHOX1* and *MOV10L1* were both expressed more in the gonocyte cluster compared to the PGC cluster (*Figure 2I*), corroborating these genes as putative markers of post-migratory PGCs across species.

## Receptor-ligand interactions between PGCs and their somatic niches across time and position

We next sought to understand how migratory PGCs interact with their somatic niche cells across time and anatomical position throughout their migration. We employed CellChat (*Jin et al., 2021*) to assess candidate intercellular signaling networks linking cell types in anterior and posterior positions at E9.5 and E10.5. At E9.5, posterior PGCs remain exclusively within the hindgut, while the most advanced anterior migrants have left the hindgut and are located within mesentery tissues. At E10.5, lagging migrants remain within the hindgut, the majority of PGCs are in the midst of migration in the dorsal mesentery, and the most advanced PGCs are reaching the gonadal ridges (*Cantú et al., 2016*). Despite this diversity in tissue niches, at E9.5 and E10.5, all PGCs across anterior and posterior positions received Kit signaling from their adjacent somatic cells, including the hindgut epithelium, coelomic epithelium, neural tube, and endothelial cells (*Figure 3—figure supplement 1A–D*). Kit signaling information flow was not found to be significantly different between anterior and posterior

migrants at E9.5 or E10.5 (*Figure 3A and B*). This is consistent with prior reports that PGCs are surrounded by KitL-expressing cells throughout their entire migration (*Gu et al., 2009*).

Across several signaling pathways, we identified dramatic cell-cell communication changes between developmental timepoints and posterior and anterior tissue niches. One major finding was significant ncWnt signaling unique to posterior PGC migrants; the source of this signaling was the hindgut and coelomic epithelia at E9.5 and primarily the coelomic epithelium at E10.5 (*Figure 3C*). This corroborates earlier experimental work demonstrating ncWNT signaling in the hindgut that suppresses PGC proliferation while in this early migratory niche (*Cantú et al., 2016*). The hindgut and coelomic epithelium-mediated ncWNT signaling to PGCs was not present in the anterior migrants at E9.5 (data not shown) or E10.5 (*Figure 3D and E*), suggesting a transient and important (*Laird et al., 2011*) developmental signaling niche specific to posterior migrants. In further support, our reanalysis of signaling networks in human migratory PGCs (*Li et al., 2017*) revealed that ncWNT signaling from the hindgut epithelium to early migratory PGCs is conserved in human embryos (*Figure 3F*).

At E9.5, we found that FGF signaling was significantly enriched in the anterior cells compared to the posterior (*Figure 3A*). In addition, the source of FGF shifted from the hindgut and neural lineages in the posterior to primarily the coelomic epithelium in the anterior (*Figure 3—figure supplement 1A, B, and E*). FGF signaling is involved in germ cell homing in zebrafish (*Chang et al., 2020*) and promotes survival and proliferation in migrating avian (*Whyte et al., 2015*) and mouse PGCs (*Takeuchi et al., 2005*). The niche-specific shift in FGF signaling source and intensity may support a shift to greater proliferation following hindgut exit in anterior migrants (*Cantú et al., 2016*).

Human PGCs have been shown to use nerve fibers as a migration substrate according to one report (*Møllgård et al., 2010*). Although mouse germ cells have not been shown to colocalize with nerve fibers (*Wolff et al., 2019*), we identified NCAM as an enriched signaling pathway (*Figure 3A*) from neural tube, neural progenitors, hindgut epithelium, and coelomic epithelium within the mesentery migratory niche. NCAM signaling to PGCs was only found in anterior migrants at E9.5 (*Figure 3—figure supplement 1F*) and was enriched in anterior migrants over posterior at E10.5 (*Figure 3A and B*).

We also observed Eph/Ephrin signaling as an enriched cell communication pathway in both the human and mouse datasets; Ephrins have established roles in coordination of adhesion and motility (*Santiago and Erickson, 2002*; *Singh et al., 2012*) and were recently identified as relevant for germ cell development in *Xenopus* (*Owens et al., 2017*) and mouse (*Alexander et al., 2024*). The spatial resolution in our mouse data reveals that posterior E9.5 PGCs exclusively send Ephrin type A signals to the hindgut epithelium (*Figure 3—figure supplement 1G*) while receiving Ephrin type B from the coelomic epithelium (*Figure 3—figure supplement 1H*); anterior E9.5 PGCs both send and receive Ephrins type A and B to and from the coelomic epithelium (*Figure 3—figure supplement 1G and H*). E10.5 posterior PGCs continue to send and receive Ephrin type A signaling to and from the coelomic epithelium, but shift to receive Ephrin type B from endothelial cells and neural progenitors in the mesentery (*Figure 3—figure supplement 1I and J*). Notably, E10.5 anterior PGCs located at the final migration station before entering the gonads do not participate in Ephrin A or B signaling as either senders or receivers (*Figure 3—figure supplement 1I and J*). Human migratory PGCs receive Ephrins type A from the mesenchyme and endothelial cells and receive Ephrins type B from immune populations (*Figure 3G and H*); once they reach the gonad as gonocytes, these cells continue to receive Ephrin B from the same sources, as well as the gonadal soma, but instead send Ephrin type A to epithelial, endothelial, mesenchymal, and immune populations (*Figure 3G*). These data suggest a role for Ephrins in modulating distinct cell-cell interactions between PGCs and their diverse migratory niches.

## Transcriptomic shifts across anatomical position in E9.5 migratory PGCs

The E9.5 anterior PGC population was comprised of leading migrants situated in the more developmentally advanced mesentery niche, as well as migrants in the anterior portions of the hindgut. Conversely, the E9.5 posterior PGC population encompassed migrants located mainly within the hindgut, occupying just the initial migratory niche. To identify transcriptomic differences by migratory position, we analyzed E9.5 PGCs alone. Initial clustering analysis revealed strong co-clustering of cells by cell cycle status (*Figure 4—figure supplement 1A*); thus, we regressed out cell cycle phase based on S and G2M scores to reveal additional sources of variation. The resultant cell embedding revealed

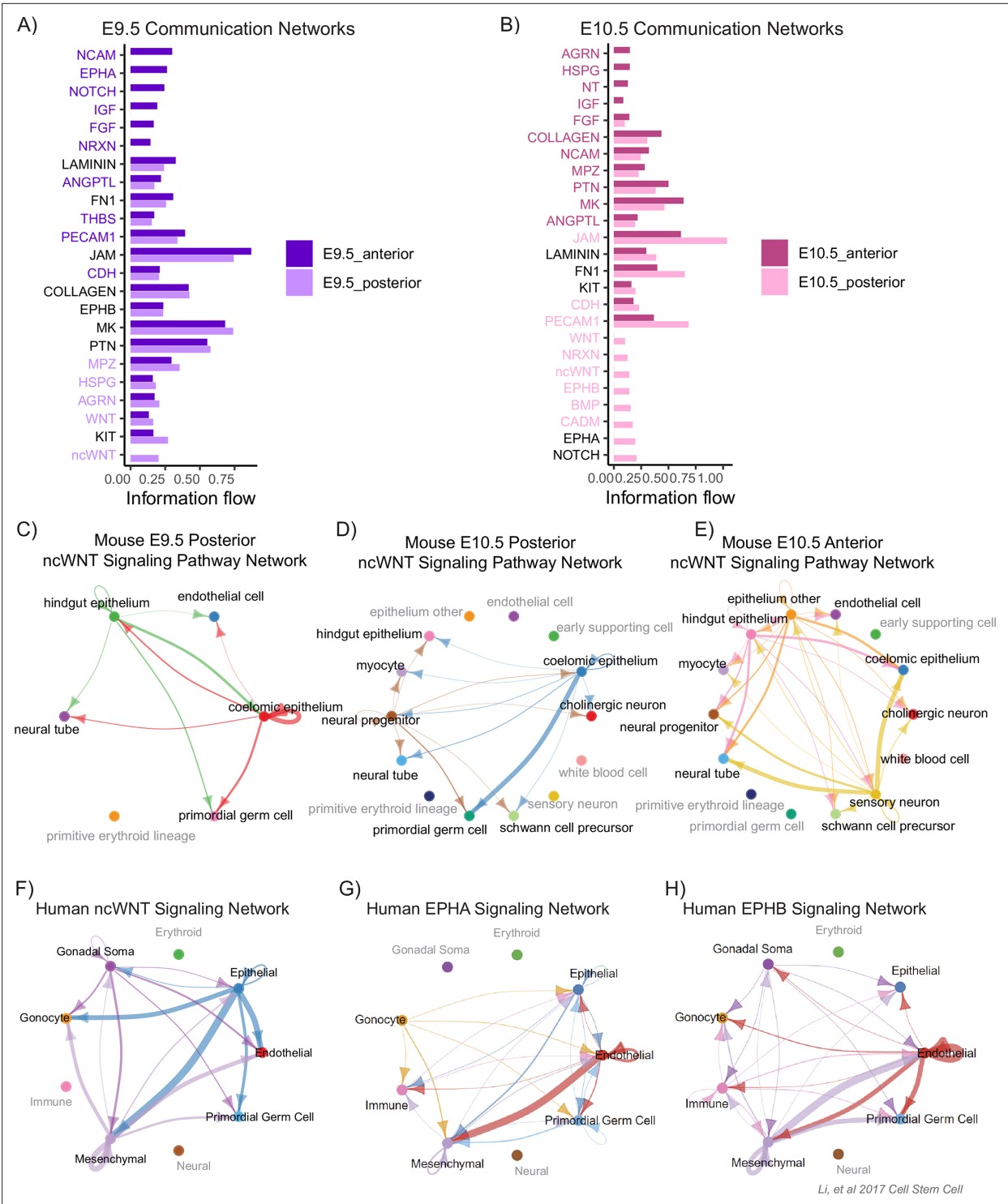

**Figure 3.** Cell signaling dynamics reveal position-dependent pathways in migrating mouse and human primordial germ cells. (**A**) Comparison of intercellular communication networks enriched in E9.5 anterior cells vs enriched in E9.5 posterior cells by normalized information flow computed with CellChat. (**B**) Comparison of intercellular communication networks enriched in E10.5 anterior cells vs enriched in E10.5 posterior cells computed with CellChat. In A and B, the color of the signaling pathway name corresponds to whether it is significantly enriched in either anterior or posterior. Black pathway names are not significantly differentially enriched. Colors correspond to the legend. (C–H) Cell populations in gray text do not participate significantly in plotted signaling pathway based on CellChat analysis. Arrows are colored according to cell type from which they originate. (**C**) Non-canonical WNT (ncWNT) signaling pathway network within mouse E9.5 posterior. (**D**) ncWNT signaling pathway network within mouse E10.5 posterior.

*Figure 3 continued on next page*

*Figure 3 continued*

(**E**) ncWNT signaling pathway network within mouse E10.5 anterior. (**F**) Human ncWNT signaling pathway network. (**G**) Human Ephrin A signaling pathway network. (**H**) Human Ephrin B signaling pathway network.

The online version of this article includes the following figure supplement(s) for figure 3:

**Figure supplement 1.** Position-dependent signaling networks between mouse primordial germ cells and their somatic niches.

that posterior cells predominantly clustered together, with a transition zone leading to two discrete zones of predominantly anterior cells (*Figure 4A*).

To probe transcriptional heterogeneity between anterior and posterior migrants, we conducted differential gene expression on pseudobulked anterior vs posterior populations with pseudoreplicates. We found that Thymosin-β4 (*Tmsb4x*), a globular G-actin-binding protein previously implicated in planar cell polarity (*Padmanabhan et al., 2020*), and *Tpm4*, a member of the tropomyosin family of actin binding proteins implicated in cell motility and adhesion (*Jeong et al., 2017*; *Zhao et al., 2019*), were both upregulated in posterior PGCs relative to anterior (*Figure 4B*). Using CellRank's pseudo-time kernel on the E9.5 anterior and posterior samples, we identified one initial and four terminal states (*Figure 4C*) with similar probability (*Figure 4D*). Transcripts that differ between anterior and posterior PGCs both decrease in expression uniformly in inferred differentiation trajectories leading to terminal states among E9.5 anterior PGCs (*Figure 4E*). GSEA on differentially expressed genes across

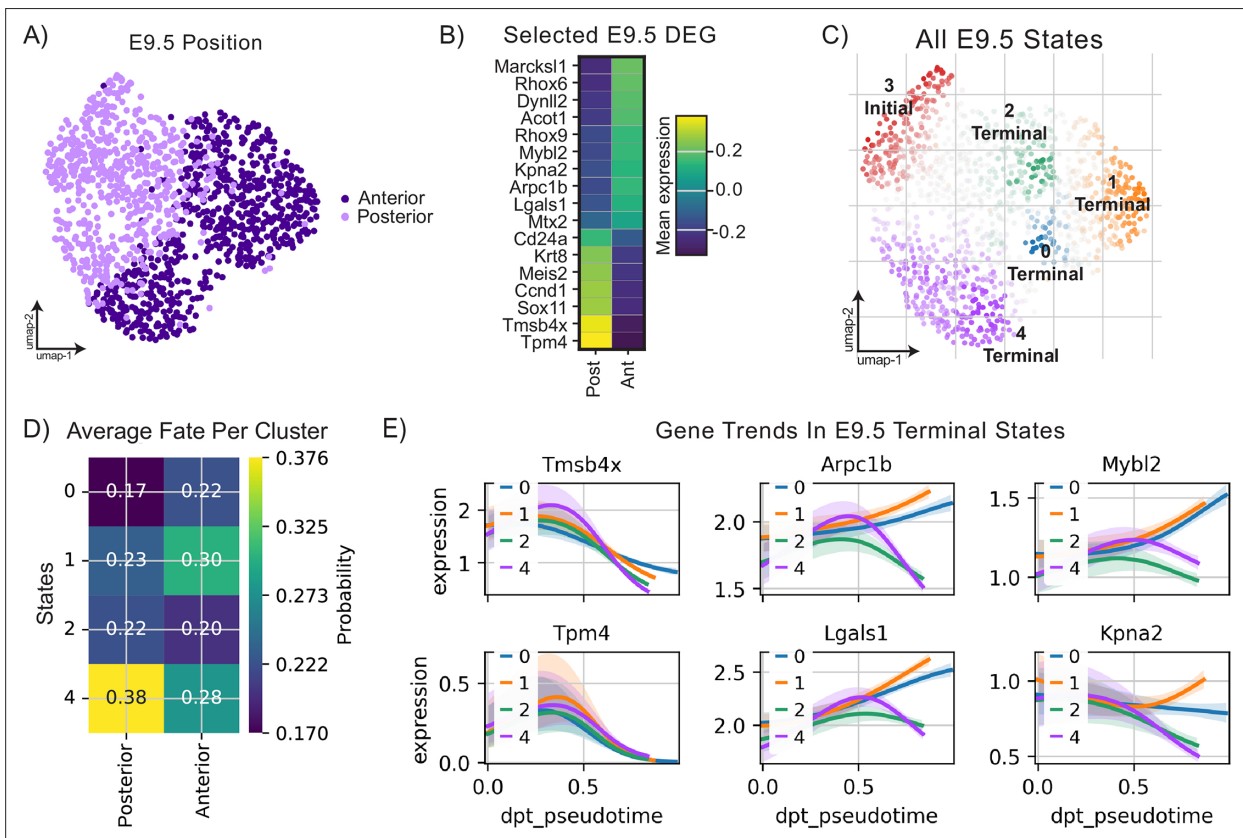

**Figure 4.** Analysis of migratory mouse primordial germ cells at E9.5 reveals transcriptional programs and putative differentiation trajectories. (**A**) Uniform Manifold Approximation and Projection (UMAP) plot of E9.5 primordial germ cells (PGCs) colored by anteroposterior position. (**B**) Matrixplot of expression of selected differentially expressed genes between anterior and posterior E9.5 PGCs. (**C**) All initial and terminal states predicted from CellRank transition matrix analysis computed with diffusion pseudotime plotted on UMAP representation of E9.5 PGCs. (**D**) Matrixplot of fate probability of reaching identified terminal states by position identity of PGCs at E9.5. (**E**) Gene expression trends between initial state 3 and the four identified terminal states for selected genes.

The online version of this article includes the following figure supplement(s) for figure 4:

**Figure supplement 1.** Regression of cell cycle phase reveals position dependant gene ontology analysis and putative differentiation trajectory driver genes in E9.5 mouse primordial germ cells.

anatomical position revealed several terms related to cell-matrix interactions relevant for migration (*Figure 4—figure supplement 1B*). Glycosaminoglycans and proteoglycans have been previously shown to exhibit location-specific distributions along the PGC migratory path (*Soto-Suazo et al., 2002*) these data raise the possibility that distinct niche interactions modulate PGC transcriptomes through changes in glycosaminoglycan biosynthesis and degradation (*Figure 4—figure supplement 1B*), which may in turn modulate migratory behavior across anatomical position.

We hypothesize that E9.5 anterior PGCs modulate expression of their actin polymerization machinery and cell surface-extracellular matrix interaction molecules as they transition from a chiefly migratory to more proliferative phenotype with distinct adhesion properties. Anterior PGCs at this timepoint upregulate *Arpc1b*, a member of the Arp2/3 complex (*Laurila et al., 2009*; *Gamallat et al., 2022*; *Rauhala et al., 2013*), alongside *Lgals1*, known to promote or inhibit collective cell migration depending on the signaling context by upregulating downstream integrins (*Auvynet et al., 2013*; *Rizqiawan et al., 2013*; *Figure 4B*). Interestingly, these genes exhibit diverging gene expression trends among identified terminal states (*Figure 4E*). Transcriptomic shifts between leading and lagging migrants extend beyond cytoskeletal regulation and extracellular matrix interactions. *Mybl2*, a potent pro-proliferative and pro-survival factor (*Musa et al., 2017*), is also upregulated in E9.5 anterior migrants, reflecting the shift to increased proliferation as PGCs encounter niches outside the hindgut. Finally, *Kpna2*, a gene encoding an importin previously shown to be required for spermatogenesis (*Navarrete-López et al., 2023*), was upregulated in anterior E9.5 PGCs (*Figure 4B*) and was found as a driver gene for one of four identified terminal states among anterior PGCs (*Figure 4—figure supplement 1C*). *Kpna2* and *Mybl2* also exhibit divergent trends in gene expression across terminal states, suggesting their potential involvement in alternative differentiation trajectories among early migrants.

## Transcriptomic shifts across anatomical position in E10.5 migratory PGCs

After cell cycle phase regression (*Figure 5—figure supplement 1A*), anterior and posterior E10.5 PGCs clustered mostly with cells of shared positional origin, with some cells appearing to occupy an intermediate between anterior and posterior states (*Figure 5A*). Differential gene expression analysis of anterior vs posterior E10.5 PGCs revealed genes previously implicated in fertility relevant to spermatogenesis and oogenesis. Genes upregulated in anterior migrants included *Kpna2*, which remained upregulated at E10.5 to a similar degree as at E9.5, *Zfp42 (Rex1)*, an epigenetic regulator of genomic imprinting (*Kim et al., 2011*) and transcription factor necessary for proper differentiation of early and late spermatids (*Rezende et al., 2011*), and *Mpc1*, one of two mitochondrial pyruvate transporters important for early ovarian folliculogenesis (*Tanaka et al., 2021*; *Figure 5B*).

We found *Lefty1* and *Lefty2* among the top differentially expressed genes upregulated in the E10.5 anterior cells compared to posterior (*Figure 5B*). Lefty mRNA upregulation is a direct response to Nodal signaling (*Shen, 2007*), so we sought to identify candidate sources of Nodal ligand. Surveying Nodal expression across somatic and germ cells across timepoints revealed strong Nodal expression in PGCs from E10.5 (*Figure 5—figure supplement 1B*). Taken together, these results suggest differences in autocrine Nodal signaling across migratory spatial position at E10.5. A known target of Nodal signaling and mitochondrial gene involved in proline synthesis *Pycr2* (*Lee et al., 2011*) was also found to be significantly upregulated in anterior PGC migrants. Additionally, after conducting GO analysis on differentially expressed genes upregulated in anterior E10.5 PGCs relative to posterior, *Nodal* signaling terms were enriched (*Figure 5—figure supplement 1C*). Nodal signaling was previously implicated in collective cell migration, with increased cell protrusion and internalization movements correlated with high Nodal signaling activity (*Pinheiro et al., 2022*) partly through induction of *Pitx2* (*Collins et al., 2018)*, which was also upregulated in anterior migrants at E10.5 (*Figure 5B*). A recent preprint also found that Nodal influences cardiac progenitor cell migration by modulating F-actin activity (*Gonzalez et al., 2025*). *Fscn1*, which encodes an actin bundling protein, was significantly more highly expressed in anterior migrants than posterior migrants at E10.5 (*Figure 5B*). This gene is a direct target gene of Nodal signaling and is required to facilitate nuclear localization of phospho-Smad2 by shuttling internalized Nodal-Activin responsive TGF-beta type 1 receptors to early endosomes (*Liu et al., 2016*). In addition, *Nodal* upregulation in late migratory PGCs may be conserved in human embryos, as hPGCs express *NODAL* most highly from 9 to 10 W (*Figure 5—figure supplement 1D*). hPGCs

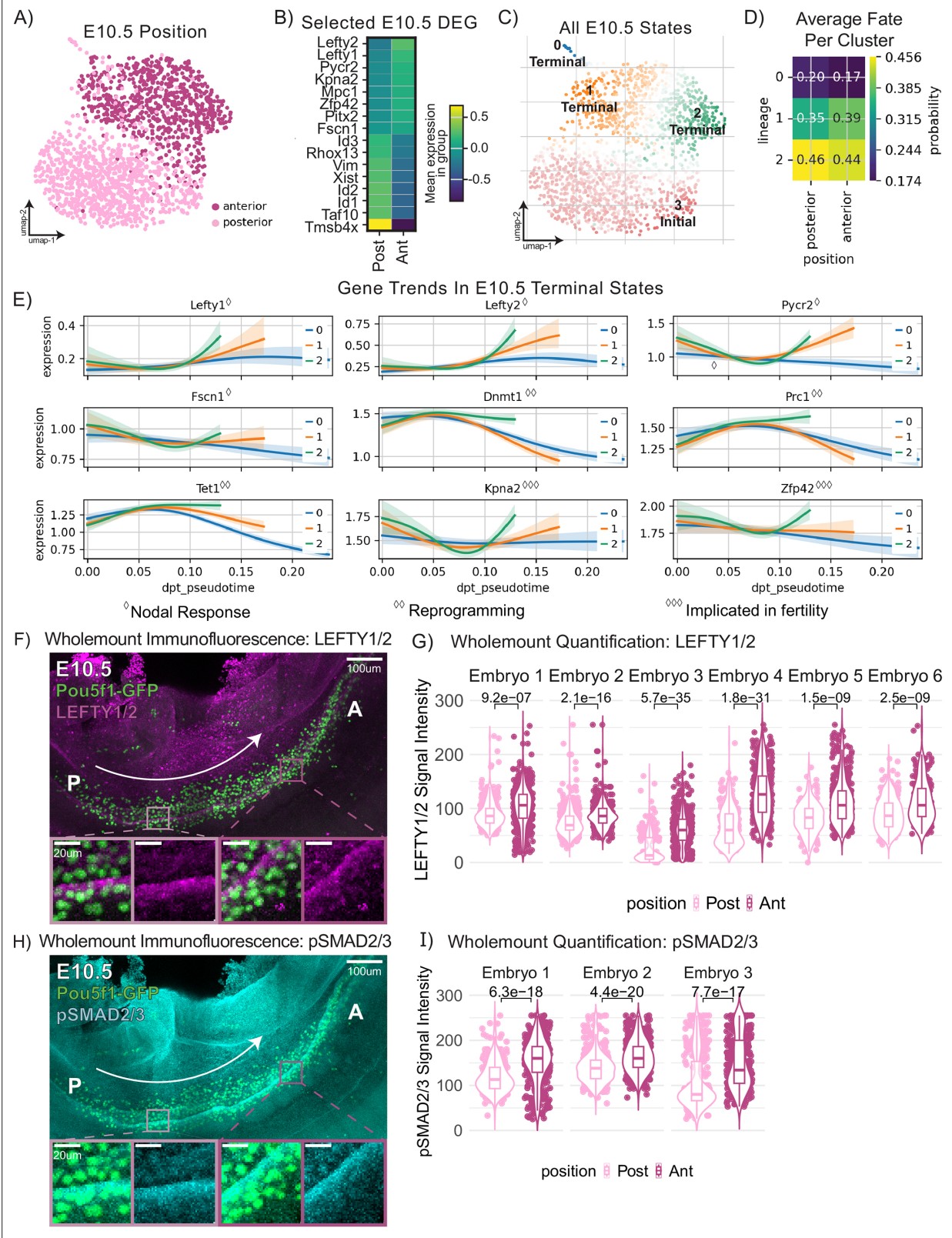

**Figure 5.** Differential expression of LEFTY1/2 and SMAD2/3 across E10.5 migratory mouse primordial germ cells defines distinct developmental states. (**A**) Uniform Manifold Approximation and Projection (UMAP) plot of E10.5 primordial germ cells (PGCs) colored by anteroposterior position. (**B**) Matrixplot of expression of selected differentially expressed genes between anterior and posterior E10.5 PGCs. (**C**) All initial and terminal states predicted from CellRank transition matrix analysis computed with diffusion pseudotime plotted on UMAP representation of E10.5 PGCs. (**D**) Matrixplot

*Figure 5 continued on next page*

*Figure 5 continued*

of fate probability of reaching identified terminal states by position identity of PGCs at E10.5. (**E**) Gene expression trends between initial state 3 and the four identified terminal states for selected genes. Diamonds correspond to functional relevance of selected genes. (**F**) Representative image of whole-mount immunofluorescence staining of Oct4-ΔPE-eGFP (Oct4GFP) and LEFTY1/2. Fluorescence from Oct4GFP reporter signal is amplified by anti-GFP antibody (ab13970) stain. For both the whole image and zoom-in portion, we selected slices for the maximum projection based on the presence of GFP signal. (**G**) Quantification of LEFTY1/2 signal intensity of PGCs in n=6 embryos, one of which is shown in F. (**H**) Representative image of whole-mount immunofluorescence staining of Oct4GFP and pSMAD2/3. Fluorescence from Oct4-ΔPE-eGFP reporter signal is amplified by anti-GFP antibody (ab13970) stain. For both the whole image and zoom-in portion, we selected slices for the maximum projection based on the presence of GFP signal. (**I**) Quantification of pSMAD2/3 signal intensity of PGCs in n=3 embryos, one of which is shown in F. p-Values for all quantifications were conducted using a paired Wilcoxon signed-rank test with BH adjustment in R.

The online version of this article includes the following figure supplement(s) for figure 5:

**Figure supplement 1.** Differential expression and differentiation trajectory-associated driver genes highlight Nodal-Lefty signaling differences over time in mouse and human primordial germ cells.

---

express *LEFTY1*, then downregulate its expression by the time they become gonocytes in the early gonad (*Figure 2—figure supplement 1G*), suggesting a similar developmental time course of this signaling axis in human embryos.

We next constructed and assessed CellRank trajectories to identify putative drivers of terminal states within the E10.5 timepoint alone. We specified the root cell to originate from the E10.5 posterior population and found a single initial state composed mostly of posterior cells. We found two terminal states (orange and green, *Figure 5C*), primarily consisting of anterior cells and one terminal state (blue, *Figure 5C*) consisting of a mix of anterior and posterior cells. Plotting terminal state probabilities by position revealed that many posterior PGCs were assigned high fate probabilities (*Figure 5D*) of reaching terminal state 2, suggesting a commonly traversed differentiation pathway linking the initial state with terminal state 2. This was supported by the finding that genetic drivers of terminal state 2 included genes involved in PGC epigenetic reprogramming and survival: *Tet1* (*Hill et al., 2018*), *Prc1* (*Mochizuki et al., 2021*), *Chd4* (*Zhao et al., 2017*; *de Castro et al., 2022*), *Dnmt1* (*Hargan-Calvopina et al., 2016*), and *Zcwpw1* (*Yuan et al., 2022*; *Wells et al., 2020*; *Figure 5—figure supplement 1E*). We previously showed that later germ cells exhibiting inappropriately prolonged DNA methylation in the fetal testis are prone to later elimination by apoptosis during sex differentiation (*Nguyen et al., 2020*). The high representation of reprogramming-related genes as drivers of terminal state 2 therefore suggests that this state is consistent with appropriate reprogramming and possibly future survival in the developing germline. In addition, we found *Mki67* as a driver of terminal state 2 (*Figure 5—figure supplement 1E*); as PGCs in more advanced migratory niches have increased proliferative capacity (*Cantú et al., 2016*), this further supports the interpretation of terminal state 2 as a developmentally advanced cell state undergoing appropriate epigenetic reprogramming. To compare differentiation trajectories leading to the identified terminal states, we plotted trajectory-specific gene expression trends along pseudotime for key genes of interest, including genes involved in Nodal response, epigenetic reprogramming, and those with known fertility phenotypes. We found that *Lefty1* and *Lefty2* had the steepest upregulation trend within trajectory 2, but were upregulated to a lesser degree or stagnant for the other two states (*Figure 5E*). *Lefty1* and *Lefty2* may serve as markers of cells undergoing transcriptomic and epigenomic shifts consistent with differentiation into a putatively 'appropriate' terminal state 2.

To test experimentally whether increased Nodal signaling is a hallmark of anterior migrating PGCs, we performed whole-mount immunofluorescence in E10.5 embryos (*Figure 5F and H*). We found anterior E10.5 PGCs exhibited significantly increased LEFTY1/2 and nuclear phosphorylated SMAD2/3 signal intensity compared to the posterior (*Figure 5G and I*), suggesting active signaling through TGF-beta type 1 receptors that bind Nodal. These results reveal a role for Nodal signaling during the terminal phases of PGC migration, with anterior E10.5 PGCs exhibiting distinct gene expression patterns and cell-cell signaling pathways compared to posterior PGCs. Our trajectory inference analysis and immunofluorescence staining suggest that enhanced Nodal signaling in anterior PGCs is associated with successful migration and epigenetic reprogramming, providing new insights into the molecular mechanisms driving PGC development.

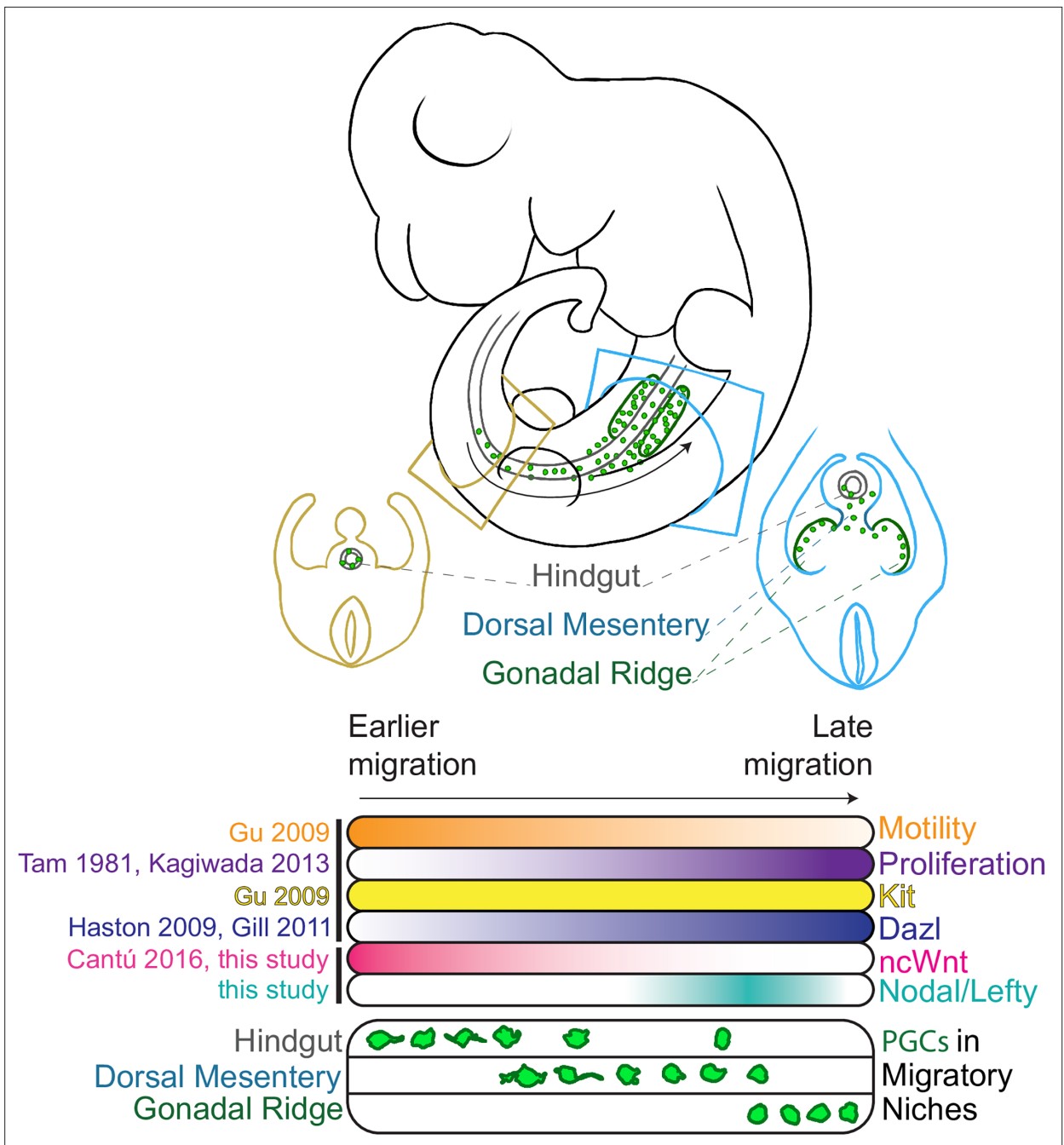

**Figure 6.** Schematic representating mouse migratory primordial germ cells and the somatic and spatial signaling cues encountered during migration. PGCs migrate through the hindgut, dorsal mesentery, and arrive in the gonadal ridge, encountering diverse signaling niches. Multiple signaling pathways, including Nodal/Lefty, non-canonical WNT (ncWnt), and survival/homing factors such as Kit, regulate PGC proliferation and migration. Seminal literature references supporting key aspects of the migratory model are listed (*Haston et al., 2009*; *Gill et al., 2011*), including contributions from this study.

## Discussion

This study provides a comprehensive view of transcriptional heterogeneity among migratory mouse PGCs and revisits an existing dataset of migratory human PGCs at analogous timepoints. Key innovations of this study include the sheer number of high-quality PGC transcriptomes profiled and separation of PGC populations by their migratory position, enabling direct comparisons between PGCs present at the same developmental timepoints but within distinct tissue niches.

One of the most striking findings of this study is the identification of Nodal signaling upregulation specific to anterior migrants at E10.5 (*Figure 6*), which we experimentally validated with whole-mount immunofluorescence. Lefty1 and Lefty2, both of which are Nodal-responsive inhibitors of Nodal, have been previously identified as part of the early PGC pluripotency network (*Seisenberger et al., 2012*), and Nodal signaling has recently been recognized as a key regulator of hPGC-like cell specification in vitro (*Jo et al., 2022*). Nodal and its co-receptor Cripto have also been identified as essential for maintaining testicular germ cell pluripotency from E12.5 to E14.5 to prevent premature spermatogenic differentiation; conversely, too much Nodal signaling during this period promotes cell invasiveness and tumor formation (*Spiller et al., 2012*). At E13.5, *Lefty2* is a marker of putatively developmentally defective apoptosis-poised testicular germ cells that have not undergone differentiation to prospermatogonia (*Nguyen et al., 2020*). Importantly, we observed no obvious correlation between a cell's *Lefty1* or *Lefty2* expression and its Y-chromosome transcript expression in our E10.5 PGC dataset (*Figure 5—figure supplement 1F*), suggesting that the niche-specific Nodal signaling signature at E10.5 is unrelated to regulating testis-specific germ cell differentiation.

An additional interesting feature of this finding is that Nodal expression is not differential between anterior and posterior PGCs at E10.5, but Nodal-responsive transcripts *Lefty1, Lefty2, Pitx2, Pycr2*, and *Fscn1* are all clearly upregulated in the anterior (*Figure 5B*). This pattern could result from a transcriptional response to Nodal signaling, initiated first in leading PGC migrants, that persists after expression of the ligand has been downregulated. The Nodal-dependent upregulation of Fscn1 observed in anterior PGC migrants presents a tantalizing link between this signaling shift and changes in actin cytoskeletal dynamics associated with PGC migratory behavior (*Yamashiro et al., 1998*; *Yamakita et al., 2011*; *Lamb and Tootle, 2020*) as E10.5 anterior migrants reach the nascent gonad. Moreover, *Fscn1*'s role in potentiating TGF-beta signaling could reflect late migratory PGCs becoming primed to respond to BMP or Nodal cues once within the gonad in order to promote oogenic or spermatogenic fates, respectively (*Lundgaard Riis and Jørgensen, 2022*).

Interestingly, the initial state identified in trajectory inference analysis across PGCs from all timepoints was also identified as a terminal state (*Figure 2D*, state 2). Since state 2 is identified among E9.5 PGCs, this may reflect that some E9.5 PGCs are less able to further differentiate to later terminal states; thus, their differentiation trajectories initiate and terminate within state 2. It is known that some mouse PGCs mis-migrate and fail to reach the gonad (*Richardson and Lehmann, 2010*); PGCs with high fate probabilities for terminal state 2 may include lagging and mis-migrants. In addition, despite the fact that initial and terminal states for this analysis were based on a diffusion pseudotime ordering with a root cell identified within the E9.5 posterior sample, initial state 3 was found among primarily E9.5 anterior cells using CellRank's cell-cell transition matrix. This likely reflects that transcriptional differences between these early migrating PGC populations are relatively subtle.

Several enriched GO terms for upregulated transcripts in posterior E9.5 PGCs relative to anterior were related to heparan sulfate proteoglycan and glycosaminoglycan synthesis and degradation (*Figure 4B*). The importance of this pathway during PGC migration was further supported by our cell-cell communication analysis. Interestingly, HSPG signaling from the hindgut epithelium and coelomic epithelium in the posterior was enriched relative to anterior at E9.5. By E10.5, HSPG signaling was instead enriched in the anterior, coming primarily from endothelial cells to PGCs (*Figure 3—figure supplement 1D*). In a manner similar to Kit signaling, HSPG signaling may shift to surround PGCs as they traverse their diverse migratory niches. HSPG signaling has been previously shown to modulate migratory behavior and survival in zebrafish PGCs (*Wei and Liu, 2014*), but there has not been thorough investigation of this pathway in mouse or other mammalian systems (*García-Castro et al., 1997*). Since posterior E9.5 PGCs are mostly located within the hindgut (*Figure 1A*), the reduction in HSPG signaling in E9.5 anterior migrants may reflect changes to the PGC cytoskeleton and extracellular matrix interactions that permit hindgut exit and enable a shift toward greater proliferation in more anterior migratory niches (*Cantú et al., 2016*). If HSPG signaling indeed facilitates directional migration, reestablishment of HSPG signaling in E10.5 anterior PGCs from the endothelium may guide final PGC homing to reach the gonadal ridges.

Considering the differences observed in Ephrin signaling to PGCs between anterior and posterior niches and between mouse and human datasets (*Figure 3G and H*, *Figure 3—figure supplement 1G-J*), Ephrins likely warrant further investigation for mediating precise control of PGC migratory behavior. Ephrin B signaling mediates germ layer separation in *Xenopus* embryos; a similar mechanism

could mediate PGC repulsion from hindgut and mesentery tissues at the appropriate transitions during PGC migration. In our dataset, E10.5 anterior PGCs no longer exhibit enriched Eph/Ephrin signaling; reduced repulsive signaling between PGCs and their surrounding tissues could help PGCs bind and settle into the gonadal ridges as they complete their migration. Close analysis of the particular Eph and Ephrin ligands and receptors expressed in each somatic niche and relevant PGC subsets could help further clarify the relevance of a shifting Ephrin code in migratory PGC development.

Limitations of this study include batch effects between sequencing libraries stemming from separate library preparation procedures on cells from different timepoints and anatomical locations in our mouse dataset. When appropriate, we corrected for batch effects between libraries (see Materials and methods - Batch correction). Batch effects may have contributed some spurious differentially expressed genes between timepoints and anatomical positions, leading to false detection of some differentially regulated GO/GSEA terms and cell-cell interaction pathways. Batch effects may explain the discovery of a putative terminal differentiation state within the bulk of E9.5 anterior PGCs, as it seems unlikely that such a large proportion of the PGCs profiled at this timepoint and anatomical position represent an aberrant state that is not conducive to further PGC migration and development.

Because ectopic PGCs have been shown to give rise to germ cell tumors (*Runyan et al., 2008*), one goal of our analyses was to identify cells with clearly distinct transcriptomes consistent with mis-migration and/or aberrant differentiation. Aside from the small population of PGCs consisting of mixed anterior and posterior migrants at E10.5 (*Figure 5F*, state 0), we did not identify any minor clusters of obviously transcriptionally distinct PGCs suggesting ectopic localization. It is possible that this population is indeed composed of ectopic PGCs, but their small number precluded identification of specific markers for experimental validation. Moreover, it is possible that at the developmental stages profiled, ectopic PGCs have not yet acquired transcriptional differences reflecting their mislocalization in the embryo. Explicit profiling of ectopic PGCs separately from gonadal PGCs at E11.5 or later could be a fruitful way to establish a ground truth of expected transcriptional differences between successful and unsuccessful migrants to help identify these cells in future datasets and explore differentiation trajectories leading to the formation of germ cell tumors.

Finally, trajectory inference analyses like those in this study are more often performed on clearly divergent cell types within a differentiating lineage rather than within single cell types (*Lange et al., 2022*); thus, the differences between putative differentiation trajectories we observed at E9.5 and E10.5 may be subtler than in other applications of these methods. Overall, the abundance of PGCs profiled in this study has enabled unprecedented comparisons among PGC subtypes within and across developmental timepoints.

## Conclusion

In summary, this work provides a transcriptional survey of migrating and early post-migratory mouse and human germ cells. We interrogated migrating PGCs in the context of their somatic niche and identified known and novel cell-cell interactions that may be involved in regulation of migration. We identified transcriptional differences between PGCs in different phases along the migratory route, connecting their spatial heterogeneity to transcriptional heterogeneity. In the mouse, we assayed transcriptional differences between migratory leaders and laggards at E9.5 and E10.5, identifying increased Nodal signaling in anterior migrants at E10.5. In the human dataset, we characterized temporally specific gene expression patterns and identified key similarities and differences between migratory and post-migratory mouse and human PGCs.

## Materials and methods

### Animals

CD1 females were crossed to mixed background CD1/C57Bl6 males homozygous for Pou5f1-ΔPE-eGFP (MGI:3057158) for both tissue dissociations for single-cell RNA sequencing and whole-mount immunofluorescence validation experiments. Counting plug date as 0.5 days post conception, pregnant mice were euthanized and age-matched embryos were dissected in 0.4% bovine serum albumin (BSA) in 1x phosphate-buffered saline (PBS) at E9.5, E10.5, and E11.5. Embryos were further staged by somite number to acquire a tighter, more standardized range of developmental time for included embryos. Embryos of both sexes were pooled without genotyping, as the timepoints analyzed were

prior to sex specification. All animal work was performed under strict adherence to the guidelines and protocols set forth by the University of California San Francisco's Institutional Animal Care and Use Committee (IACUC), and all experiments were performed in an animal facility approved by the Association for the Assessment and Accreditation of Laboratory Animal Care International (AAALAC). All procedures performed were approved by the UCSF IACUC. All mice were maintained in a temperature-controlled animal facility with 12 hr light-dark cycles and were given access to food and water.

## Tissue dissociation

### E9.5 dissociation

At E9.5, five litters were dissected and embryos with total somite counts of 20–25 were kept. To split PGCs and somatic cells into anterior and posterior bins (containing migratory leaders and laggards, respectively), embryos were bisected at somite 15, corresponding to where the PGCs move from the hindgut into the dorsal mesentery (*Figure 1A*, dotted line). All tissue rostral to the forelimb was removed and the pooled anterior and posterior segments were bathed in 200 µL of pre-warmed 0.25% Trypsin/EDTA and incubated at 37°C. After 10 min, the tissue was triturated with a pipet tip enlarged by cutting with a razor blade. If after 10 min, the digest still looked clumpy, we incubated another 10 min in the 200 µL 0.25% Trypsin/EDTA. Next, 50 µL of 1 mg/mL DNase was added, samples triturated gently, and incubated for 5 min at 37°C. After 5 min, samples were checked for consistency. If they were thin, watery, and homogenous, digests were quenched with a 1:1 volume of FBS. If samples were still viscous, 50 µL more DNase was added and samples were incubated for 3 more minutes at 37°C, after which samples were quenched with a 1:1 volume of FBS and placed on ice.

### E10.5 dissociation

At E10.5, three litters were dissected and embryos with total somite counts of 34–38 were kept. To further split our E10.5 embryos into an anterior bin and posterior bin (containing migratory leaders and laggards, respectively), we did the split right below where the PGCs have begun to bifurcate to settle into the left or right gonadal ridge, between somites 19 and 20 (*Figure 1A*, dotted line). Dissected embryos were cut in half just below the heart, and the posterior half of the embryo was dissociated to prepare for FACS germ cell concentration (GFP- heads were used as GFP- control tube). Embryo's posterior halves were split into groups of 10 and placed in a 37°C water bath with 400 µL of pre-warmed 0.25% Trypsin/EDTA. An additional 200 µL of pre-warmed 0.25% Trypsin/EDTA was added if the solution appeared too thick. Embryos were digested at 37°C for 15 more minutes, and digest progress was aided by flicking and assessing solution viscosity every 5 min. 200 µL of 1 mg/mL DNase was added, samples triturated with a cut pipette (see E9.5 for description), and then incubated for 5 min in 37°C water bath. After a 5 min digest, samples were assessed for remaining clumps; if clumps remained, an additional 5 min or additional 200 µL of 1 mg/mL DNase was added. Once the solution was pipetted homogenously and water-like, it was quenched with a volume-matched amount of FBS and placed on ice.

### E11.5 AGM dissociation

At 11.5, four litters were dissected; AGMs of age-matched embryos (45–49 somite range) were micro-dissected out from posterior embryos. AGMs were grouped in sets of 6 and placed in a 37°C water bath. GFP- heads were also processed as a control. 500 µL of 0.25% Trypsin/EDTA was added to embryos. Tubes were incubated for 20 min, triturating with a cut pipet tip (see E9.5 for description) at every 5 min interval to gently mix the solution. After 20 min, 100 µL of 1 mg/mL DNase was added to each tube, and tubes were mixed by flicking. After 5 min of incubation with DNase added, solutions were checked for homogeneity. Once the solution was water-like, it was quenched with 1:1 volume of FBS and placed on ice. For the first biological replicate, 23 AGMs were dissociated. For the second replicate, 12 AGMs were dissociated.

## PGC enrichment

Once on ice, each sample was filtered through a 35 µm strainer. Tip, tube, and filter were rinsed with a small amount of 0.1% BSA to wash all possible GFP+ PGCs into the cell suspension. Sytox Blue live-dead indicator was added at the appropriate 1:1000 concentration to each digest. Samples were sorted on either a BD FACS Aria2, Aria3, or Aria Fusion, following a gating pattern as follows based on

established methods using the Pou5f1-ΔPE-eGFP mouse strain (*Spiller et al., 2017*): (1) side scatter area vs forward scatter area to select for cells; (2) side scatter width vs side scatter height to select for cell singlets; (3) forward scatter width vs forward scatter height to select again for cell singlets; (4) Sytox Blue vs forward scatter area to select for Sytox Blue negative live cells; and finally (5) GFP vs forward scatter area. The range of GFP negative vs positive gates was calibrated in each experiment based on the GFP- digest sample, to put GFP- cells in the $10^3$ range and GFP+ cells in the $10^4$–$10^5$ range. GFP+ cells were collected in 500 µL of 0.2% BSA, and GFP- somatic cells were also collected in 500 µL of 0.2% BSA. For each GPF+ library, we spiked back in somatic cells at a 1:1 ratio. We concentrated cells into a 300–1500 cells/µL range, resuspending in 0.1% BSA.

## Single-cell RNA sequencing library construction

We pooled sorted GFP+ germ cells and an equal amount of GFP- somatic cells from each timepoint and anatomical location. We loaded counted cells onto a 10X V2 chip to create a single-cell emulsion. For E9.5 anterior, we loaded 3400 cells, and for E9.5 posterior, 2800 cells. For E10.5 anterior, we loaded two technical replicates totaling 9300 cells. For E10.5 posterior, we loaded two technical replicates totaling 11,200 cells. For the two E11.5 biological replicates, we loaded 15,000 and 10,000 cells, respectively. Library creation was done in-house following 10X Chromium Single Cell 3' Reagent Kits v2 User Guide Rev A. Briefly, we used Single-Cell 3' Reagent Version 2 Kit (10X Genomics) and used fluidics to create individual cells in gel bead-in-emulsions (GEMS). Samples were prepared directly according to the protocol, with the following steps tailored to maximize library yield: at the cDNA amplification step, we used 14 cycles, and at the sample index PCR step, we used 14 cycles. Samples were tested for quality on an Agilent Bioanalyzer High Sensitivity chip before sequencing. E9.5 and E11.5 libraries were sequenced on HiSeq 4000 (Illumina) with paired-end sequencing parameters: Read1, 98 cycles; Index1, 14 cycles; Index2, 8 cycles; and Read2, 10 cycles. E10.5 libraries were sequenced on NovaSeq (Illumina) with paired-end sequencing parameters: Read1, 150 cycles; i7 Index, 8 cycles; i5 index, 0 cycles; and Read2, 150 cycles.

## Whole-mount immunofluorescence

Embryos from CD1 dams crossed to sires homozygous for Pou5f1-ΔPE-eGFP (MGI:3057158) were euthanized at E9.5 (20–25 somites) or E10.5 (34–38 somites) and then immunostained according to the iDISCO protocol (*Renier et al., 2014*) with minor modifications. Briefly, embryos were fixed in 4% PFA overnight at 4°C, blocked in 0.2% Gelatin, 0.5% Triton X-100 in 1X PBS overnight at room temperature, incubated with primary antibodies prepared in the blocking solution for 10 nights at 4°C, and incubated with secondary antibodies in the blocking solution for 2 nights at 4°C. Samples were washed in 0.2% Gelatin, 0.5% Triton X-100 at least six times for 30 min each at room temperature between incubation steps. Then, samples were cleared with 50% tetrahydrofuran (THF):dH$_2$O overnight at room temperature, 80% THF:dH$_2$O and 100% THF for 1.5 hr each at room temperature, dichloromethane for 30 min at room temperature, and finally dibenzyl ether for a minimum of 3 nights at room temperature until ready for imaging. For whole-mount immunofluorescence staining with primary antibody for pSMAD2/3, antigen retrieval was performed with 100% acetone for 1 hr at –20°C immediately before blocking. The following antibodies were used at the specified dilutions: Chicken anti-GFP (cat ab13970) 1:200, Goat anti-LEFTY1/2 (cat AF746) 1:200, Rabbit pSMAD2/3 (cat D27F4) 1:200, Donkey anti-Rat Alexa Fluor 405 (cat A48268) 1:200, Donkey anti-Chicken Alexa Fluor 488 (cat 703-546-155) 1:200, Donkey anti-Rabbit Alexa Fluor 555 (cat A31572) 1:200, Donkey anti-Goat Alexa Fluor 647 (cat A21447) 1:200.

## Imaging

All imaging was performed on the Leica TCS SP8 inverted scanning confocal microscope with 2 µm Z-steps.

## Image analysis

Image analysis to measure fluorescence intensity of LEFTY1/2 and pSMAD2/3 in E10.5 embryos was performed in Imaris software by masking on GFP signal intensity to identify PGCs expressing the Pou5f1-ΔPE-eGFP reporter and using automatic spot detection to call a 10 µm diameter spot for each PGC checked with manual correction. Then, identified PGC spots were characterized by their location

relative to a line extending ventrally from the junction of somites 19 and 20; those anterior to this location were called anterior and those posterior to this location were called posterior. Then, the max signal intensities for LEFTY1/2 or pSMAD2/3 signal were recorded, plotted, and compared with an unequal variances t-test (Welch's t-test) in Prism.

## Data processing

### SoupX decontamination

When aberrant hemoglobin gene expression in non-red blood cells was discovered, data was processed through SoupX (version 1.5.2) (*Young and Behjati, 2020*) to remove ambient RNA contamination. Contamination parameters were either set by the function autoEstCont to automatically estimate ambient RNA contamination, or set by slowly increasing the threshold of filtering until Hbb* genes ceased to inappropriately be found as markers for non-red blood cell genes, as follows: E9.5 anterior, 12%; E9.5 posterior, autoEstCont; E10.5 anterior, 19%, E10.5 posterior, autoEstCont.

Initial processing of the v2 10X libraries was done through CellRanger v3.1.1. Libraries for all timepoints were aligned to the mouse genome (10X Genomics prebuilt mm10 version 3.0.0 reference genome) with all CellRanger default parameters for demultiplexing and aligning. Our libraries were sequenced at a depth of 125,100 mean reads per cell for the E9.5 anterior sample and 111,258 mean reads per cell for the E9.5 posterior sample. At E10.5, our anterior technical replicates 1 and 2 were 493,803 and 399,673, and our posterior replicates 3 and 4 492,512 and 556,014. At E11.5, our first biologic replicate was 15,542 mean reads per cell, and the second biologic replicate 20,967. The resulting gene by cell matrices was analyzed with the Python package ScanPy v1.9.3 (*Wolf et al., 2018*). Cells were filtered to retain only high-quality cells with a minimum of 2000 transcripts per cell barcode, a maximum of 10% mitochondrial transcripts, and a doublet score of 0.15 computed with Scrublet v0.2.3 (*Wolock et al., 2019*). Further downstream, any remaining cell populations lacking unique marker genes or with a high density of doublet scores near the 0.15 doublet score cutoff were also removed.

To reanalyze the human datasets, 4–12 W count files were downloaded from GSE86146 (*Li et al., 2017*). Individual timepoints contained too few cells for Seurat v3.2.3 (*Satija et al., 2015*) to integrate effectively; therefore, male and female matrices from individual timepoints were concatenated, and the two sex-specific datasets were merged via Seurat's integration anchors-based approach. Briefly, after running NormalizeData and FindVariableFeatures to select 2000 variable genes, FindIntegrationAnchors was run (dims 1:30). ScaleData, RunPCA, and RunUMAP were then run with all default flags.

## Batch correction

To faithfully annotate cell types across timepoints, anatomical locations, and sequencing libraries, libraries were first integrated using harmonypy v0.0.9, generating a manifold that most closely co-clusters cells with transcriptomic similarity across batches to yield consistent cell-type identification across sequencing libraries. To integrate cells within timepoints but allow for visual comparison of similarities and differences between PGCs on the full cell manifold (*Figure 1C*), we used STITCH (*Wagner et al., 2018*), which joins the k-nearest neighbor graphs from multiple timepoints by identifying neighbors in adjacent timepoints and projecting cells from a timepoint $t_i$ into the next timepoint $t_{i+1}$. To handle batch variability across libraries for different timepoints, we also provided the library_id as a batch_key in order to prioritize highly variable genes identified across batches rather than batch-specific highly variable genes for use in UMAP graph construction. This light batch correction using highly variable genes was also used to build the PGC-only manifold in *Figure 2*.

## Cell-type identification

Of the cells we sequenced, 1268 E9.5 PGCs, 1664 E10.5 PGCs, and 10,330 E11.5 PGCs survived quality control filtering steps. We acquired an average of 3724 genes per cell at E9.5, 2076 genes per cell at E10.5, and 4115 genes per cell at E11.5. We next began to cluster the data and identify the germ cell and somatic cell components present. Automated cell-type label transfer was performed in CellTypist (v1.6.2) using published datasets (*Garcia-Alonso et al., 2022*; *Cao et al., 2019*) with good coverage of gonadal and extragonadal cell types at equivalent developmental stages (E9.5–11.5). Automated cell-type predictions were verified through manual gene set scoring. Clustering resolution

was varied to best capture the distribution of predicted cell types in the dataset. While the limited cell number of PGCs present in the tissue at E9.5 did not permit us to construct multiple libraries from multiple biologic replicates, we prepared the E10.5 anterior and posterior libraries, as well as the E11.5 library in biological duplicate. Automated cell-type identification was performed on the *Li et al., 2017*, human dataset using CellTypist's Developing_Human_Gonads model based on *Garcia-Alonso et al., 2022*.

### Differential expression testing

Differential expression testing was performed using pyDESeq2 (v0.4.4) and pseudobulking cells by sequencing library of origin. In cases with fewer than three libraries per timepoint or anatomical position, three pseudoreplicates were generated and cells were randomly allocated to one of these three pseudoreplicates, then gene expression data were pseudobulked by pseudoreplicate. We compared gene expression using the Wald test and corrected for multiple comparisons via the Benjamini-Hochberg procedure.

### Gene set enrichment analysis

GSEA was performed with GSEApy (v1.1.2) on differentially expressed genes from pyDESeq2 (see Differential expression testing) filtered to include genes with a log2 fold change ≥0.5 or ≤–0.5 and an adjusted p-value of ≤0.01. Gene sets used for enrichment analysis were acquired from the Molecular Signatures Database (MSigDB).

### Trajectory inference methods

We used CellRank (v2.0.2) (*Lange et al., 2022*) to analyze differentiation trajectories across and within timepoints. We began by using CellRank's real-time kernel with germ cells from E9.5 to E11.5 to generate a cell-cell transition matrix incorporating transcriptional similarities among germ cells, as well as their developmental stage, subsampling the data to contain the same number of germ cells from each timepoint to reduce biases from cell number. Assigning E9.5 posterior to contain the starting state, we plotted random walks through the transition matrix to identify the most likely endpoints of differentiation. Then, using diffusion pseudotime computed in ScanPy for cells across all timepoints in CellRank's pseudotime kernel, we identified putative initial and terminal macrostates within migratory PGCs. For all trajectory analyses, the number of macrostates used was the lowest possible such that the CellRank algorithm identified an initial state containing the root cell used for pseudotemporal ordering. Each cell not assigned to a macrostate was scored on its likelihood of reaching a predicted terminal state using Markov chain absorption probabilities, which approximate the results of infinite random walks through the state manifold. For intra-timepoint trajectory analyses at E9.5 and E10.5, we used CellRank's pseudotime kernel with diffusion pseudotime computed in ScanPy to build cell-cell transition matrices. Intra-timepoint initial and terminal states, as well as random walks, were also computed within CellRank's pseudotime kernel based on diffusion pseudotime.

### Cell-cell communication analysis

All cell-cell communication network analysis was performed in CellChat (v2.1.2) (*Jin et al., 2021*). Enriched cell-cell communication networks were initially inferred from somatic and germ cells derived from each timepoint and anatomical position separately (E9.5 posterior, E9.5 anterior, E10.5 posterior, and E10.5 anterior), and then CellChat objects were compared via cross-dataset analysis. Since initial cell-type annotations were performed on the integrated cell manifold across all cells from E9.5 to E11.5, subsetted embeddings for each position and timepoint were checked for annotation accuracy and cluster annotations were slightly modified as needed prior to CellChat analysis.

## Acknowledgements

The authors would like to acknowledge Dan Bunis, David Wu, Aparna Bhadui, Bikem Soygur, Dan Nguyen, Lina Afonso, and Anjali Prabhu for their support of this work through helpful discussions and contributions to data collection and analysis. We thank the San Francisco Chan Zuckerberg Biohub for sequencing support. This work was funded by NIH grants 1DP2OD007420 (DJL), 1R01GM122902 (DJL), 1R01ES023297 (DJL), 1F31HD096840-01 (RGJ), 1F30HD117592 (JWZ), as well as the San

Francisco Chan-Zuckerberg Biohub (DJL, DEW) and Global Consortium for Reproductive Health through the Bia-Echo Foundation GCRLE-0123 (DJL).

## Additional information

### Competing interests

Diana J Laird: Scientific advisor at Vitra, Inc. The other authors declare that no competing interests exist.

### Funding

| Funder | Grant reference number | Author |
|---|---|---|
| National Institutes of Health | 1DP2OD007420 | Diana J Laird |
| National Institutes of Health | 1R01GM122902 | Diana J Laird |
| National Institutes of Health | 1R01ES023297 | Diana J Laird |
| National Institutes of Health | 1F31HD096840 | Rebecca Garrett Jaszczak |
| Chan Zuckerberg Initiative | | Daniel E Wagner Diana J Laird |

The funders had no role in study design, data collection and interpretation, or the decision to submit the work for publication.

### Author contributions

Rebecca Garrett Jaszczak, Investigation, Writing – original draft, Writing – review and editing, Data curation, Formal analysis; Jay W Zussman, Data curation, Formal analysis, Investigation, Writing – original draft, Writing – review and editing; Daniel E Wagner, Supervision, Methodology, Writing – review and editing; Diana J Laird, Conceptualization, Supervision, Funding acquisition, Project administration, Writing – review and editing

### Author ORCIDs

Rebecca Garrett Jaszczak ⓘ https://orcid.org/0000-0003-2588-8790
Jay W Zussman ⓘ https://orcid.org/0000-0003-4727-0289
Daniel E Wagner ⓘ https://orcid.org/0000-0002-2983-635X
Diana J Laird ⓘ https://orcid.org/0000-0002-4930-0560

### Ethics

This study was performed in strict accordance with the recommendations in the Guide for the Care and Use of Laboratory Animals of the National Institutes of Health. All of the animals were handled according to approved institutional animal care and use committee (IACUC) protocols of the University of California San Francisco, AAALAC approval #001084 and Public Health Service (PHS) Animal Welfare Assurance number D16-00253/A3400-01.

Reviewer #1 (Public review): https://doi.org/10.7554/eLife.103074.3.sa1
Reviewer #2 (Public review): https://doi.org/10.7554/eLife.103074.3.sa2
Reviewer #3 (Public review): https://doi.org/10.7554/eLife.103074.3.sa3
Author response https://doi.org/10.7554/eLife.103074.3.sa4

## Additional files

### Supplementary files

MDAR checklist

## Data availability

Raw and processed transcriptomic data generated in this study have been deposited in the Gene Expression Omnibus (GEO) under the accession code GSE274603. Code used to analyze the data is located at https://github.com/LairdLabUCSF/Migratory-PGC-Atlas (copy archived at *Garrett Jaszczak, 2025*). Additionally, an R Shiny app to facilitate quick exploration of the mouse single cell data is located at http://rebeccagj.shinyapps.io/MousePGCAtlas.

The following dataset was generated:

| Author(s) | Year | Dataset title | Dataset URL | Database and Identifier |
|---|---|---|---|---|
| Jaszczak RG, Zussmann J, Laird DJ | 2024 | Comprehensive profiling of migratory primordial germ cells reveals niche-specific differences in non-canonical Wnt and Nodal-Lefty signaling in anterior vs posterior migrants | https://www.ncbi.nlm.nih.gov/geo/query/acc.cgi?acc=GSE274603 | NCBI Gene Expression Omnibus, GSE274603 |

The following previously published dataset was used:

| Author(s) | Year | Dataset title | Dataset URL | Database and Identifier |
|---|---|---|---|---|
| Li L | 2017 | Single-Cell RNA-Seq Analysis Maps Development of Human Germline Cells and Gonadal Niche Interactions | https://www.ncbi.nlm.nih.gov/geo/query/acc.cgi?acc=GSE86146 | NCBI Gene Expression Omnibus, GSE86146 |

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
