## [Editor Report · eLife Assessment]

This revised study provides **fundamental** insights into the differences in migratory primordial germ cells based on their anterior or posterior location. Through **convincing** methodology and analysis of single-cell RNA sequencing of an exceptionally large number of migratory primordial germ cells and surrounding somatic cells, the novel findings and datasets generated from this study provide many hypotheses of interest to germ cell biologists.

---

## [Referee Report · Reviewer #1 (Public review)]

Summary:

Migration of the primordial germ cells (PGCs) in mice is asynchronous, such that leading and lagging populations of migrating PGCs emerge. Prior studies found that interactions between the cells the PGCs encounter along their migration routes regulates their proliferation. In this study, the authors used single cell RNAseq to investigate PGC heterogeneity and to characterize their niches during their migration along the AP axis. Unlike prior scRNAseq studies of mammalian PGCs, the authors conducted a time course covering 3 distinct stages of PGC migration (pre, mid, and post migration) and isolated PGCs from defined somite positions along the AP axis. In doing so, this allowed the authors to uncover differences in gene expression between leading and lagging PGCs and their niches and to investigate how their transcript profiles change over time. Among the pathways with the biggest differences were regulators of actin polymerization and epigenetic programming factors and Nodal response genes. In addition, the authors report changes in somatic niches, specifically greater non-canonical WNT in posterior PGCs compared to anterior PGCs. This relationship between the hindgut epithelium and migrating PGCs was also detected in reanalysis of a previously published dataset of human PGCs. Using whole mount immunofluorescence, the authors confirmed elevated Nodal signaling based on detection of the LEFTY antagonists and targets of Nodal during late stage PGC migration. Taken together, the authors have assembled a temporal and spatial atlas of mouse PGCs and their niches. This resource and the data herein provide support for the model that interactions of migrating mouse PGCs with their niches influences their proliferation, cytoskeletal regulation, epigenetic state and pluripotent state.

Overall, the findings provide new insights into heterogeneity among leading and lagging PGC populations and their niches along the AP axis, as well as comparisons between mouse and human migrating PGCs. The data are clearly presented, and the text is clear and well-written. This atlas resource will be valuable to reproductive and developmental biologists as a tool for generating hypotheses and for comparisons of PGCs across species.

Strengths:

(1) High quality atlas of individual PGCs prior to, during and post migration and their niches at defined positions along the AP axis.

(2) Comparisons to available datasets, including human embryos, provide insight into potentially conserved relationships among PGCs and the identified pathways and gene expression changes.

(3) Detailed picture of PGC heterogeneity.

(4) Valuable resource for the field.

(5) Some validation of Nodal results and further support for models in the literature based on less comprehensive expression analysis.

---

## [Referee Report · Reviewer #2 (Public review)]

Summary:

Germ cells go on to form sperm and eggs and are, therefore, critical for the survival of the species. This work addresses the question of how 'leading' and 'lagging' PGCs differ, molecularly, during their migration to the mouse genital ridges/gonads during fetal life (E9.5, E10.5, E11.5), and how this is regulated by different somatic environments encountered during the process of migration. E9.5 and E10.5 cells differed in expression of genes involved in canonical WNT signaling and focal adhesions. Differences in cell adhesion, actin cytoskeletal dynamics were identified between leading and lagging cells, at E9.5, before migration into the gonads. At E10.5, when some PGCs have reached the genital ridges, differences in Nodal signaling response genes and reprogramming factors were identified. This last point was verified by whole mount IF for proteins downstream of Nodal signaling, Lefty1/2. At E11.5, there was upregulation of genes associated with chromatin remodeling and oxidative phosphorylation. Some aspects of the findings were also found to be likely true in human development, established via analysis of a dataset previously published by others.

Strengths:

The work is strong in that a large number of PGCs were isolated and sequenced, along with associated somatic cells. The authors dealt with the problem of a very small number of migrating mouse PGCs by pooling cells from embryos (after ascertaining age matching using somite counting). 'Leading' and 'lagging' populations were separated by anterior and posterior embryo halves and the well-established Oct4-deltaPE-eGFP reporter mouse line was used.

The most likely possible use of this fundamental information will be the incorporation of some aspects (e.g. the potential importance of Nodal signaling) into protocols for generation of in vitro derived gametes.

---

## [Referee Report · Reviewer #3 (Public review)]

Summary:

The migration of primordial germ cells (PGCs) to the developing gonad is a poorly understood yet essential step in reproductive development. Here, the authors examine whether there are differences in leading and lagging migratory PGCs using single-cell RNA sequencing of mouse embryos. Cleverly, the authors dissected embryonic trunks along the anterior-to-posterior axis prior to scRNAseq in order to distinguish leading and lagging migratory PGCs. After batch corrections, their analyses revealed several known and novel differences in gene expression within and around leading and lagging PGCs, intercellular signaling networks, as well as number of genes upregulated upon gonad colonization. The authors then compared their datasets with publicly available human datasets to identify common biological themes. Altogether, this rigorous study reveals several differences between leading and lagging migratory PGCs, hints at signatures for different fates among the population of migratory PGCs, and provides new potential markers for post-migratory PGCs in both humans and mice. While many of the interesting hypotheses that arise from this work are not extensively tested, these data provide a rich platform for future investigations.

Strengths:

The authors have successfully navigated significant technical challenges to obtain a substantial number of mouse migratory primordial germ cells for robust transcriptomic analysis. Here, the authors were able to collect quality data on ~13,000 PGCs and ~7,800 surrounding somatic cells, which is ten times more PGCs than previous studies.

The decision to physically separate leading and lagging primordial germ cells was clever and well-validated based on expected anterior-to-posterior transcriptional signatures.

Within the PGCs and surrounding tissues, the authors found many gene expression dynamics they would expect to see both along the PGC migratory path as well as across developmental time, increasing confidence in the new differentially expressed genes they found.

The comparison of their mouse-based migratory PGC datasets with existing human migratory PGC datasets is appreciated.

The quality control, ambient RNA contamination elimination, batch correction, cell identification and analysis of scRNAseq data were thorough and well-done such that the new hypotheses and markers found through this study are dependable.

The subsetting of cells in their trajectory analysis is appreciated, further strengthening their cell terminal state predictions.

Weaknesses:

There were a few validation experiments within this study. For one such experiment, whether there is a difference in pSMAD2/3 along the AP axis is unclear and not quantified, as was nicely done for Lefty1/2.

---

## [Author Response]

The following is the authors’ response to the original reviews.

**Reviewer #1 (Public review):**
Summary:Migration of the primordial germ cells (PGCs) in mice is asynchronous, such that leading and lagging populations of migrating PGCs emerge. Prior studies found that interactions between the cells the PGCs encounter along their migration routes regulates their proliferation. In this study, the authors used single cell RNAseq to investigate PGC heterogeneity and to characterize their niches during their migration along the AP axis. Unlike prior scRNAseq studies of mammalian PGCs, the authors conducted a time course covering 3 distinct stages of PGC migration (pre, mid, and post migration) and isolated PGCs from defined somite positions along the AP axis. In doing so, this allowed the authors to uncover differences in gene expression between leading and lagging PGCs and their niches and to investigate how their transcript profiles change over time. Among the pathways with the biggest differences were regulators of actin polymerization and epigenetic programming factors and Nodal response genes. In addition, the authors report changes in somatic niches, specifically greater non-canonical WNT in posterior PGCs compared to anterior PGCs. This relationship between the hindgut epithelium and migrating PGCs was also detected in reanalysis of a previously published dataset of human PGCs. Using whole mount immunofluorescence, the authors confirmed elevated Nodal signaling based on detection of the LEFTY antagonists and targets of Nodal during late stage PGC migration. Taken together, the authors have assembled a temporal and spatial atlas of mouse PGCs and their niches. This resource and the data herein provide support for the model that interactions of migrating mouse PGCs with their niches influences their proliferation, cytoskeletal regulation, epigenetic state and pluripotent state.Overall, the findings provide new insights into heterogeneity among leading and lagging PGC populations and their niches along the AP axis, as well as comparisons between mouse and human migrating PGCs. The data are clearly presented, and the text is clear and well-written. This atlas resource will be valuable to reproductive and developmental biologists as a tool for generating hypotheses and for comparisons of PGCs across species.Strengths:(1) High quality atlas of individual PGCs prior to, during and post migration and their niches at defined positions along the AP axis.(2) Comparisons to available datasets, including human embryos, provide insight into potentially conserved relationships among PGCs and the identified pathways and gene expression changes.(3) Detailed picture of PGC heterogeneity.(4) Valuable resource for the field.(5) Some validation of Nodal results and further support for models in the literature based on less comprehensive expression analysis.Weaknesses:(1) No indication of which sex(es) were used for the mouse data and whether or not sex-related differences exist or can excluded at the stages examined. This should be clarified.

We have added: “Embryos of both sexes were pooled without genotyping, as the timepoints analyzed were prior to sex specification” to both the Animals section of the Materials and Methods and the Figure 1 legend. In addition, bioinformatic evaluation of potential sex biases in Nodal-Lefty signaling using Y-chromosome gene expression is reported in supplementary figure 4 and discussed in Discussion paragraph 2.

**Reviewer #2 (Public review):**
Summary:This work addresses the question of how 'leading' and 'lagging' PGCs differ, molecularly, during their migration to the mouse genital ridges/gonads during fetal life (E9.5, E10.5, E11.5), and how this is regulated by different somatic environments encountered during the process of migration. E9.5 and E10.5 cells differed in expression of genes involved in canonical WNT signaling and focal adhesions. Differences in cell adhesion, actin cytoskeletal dynamics were identified between leading and lagging cells, at E9.5, before migration into the gonads. At E10.5, when some PGCs have reached the genital ridges, differences in Nodal signaling response genes and reprogramming factors were identified. This last point was verified by whole mount IF for proteins downstream of Nodal signaling, Lefty1/2. At E11.5, there was upregulation of genes associated with chromatin remodeling and oxidative phosphorylation. Some aspects of the findings were also found to be likely true in human development, established via analysis of a dataset previously published by others.Strengths:The work is strong in that a large number of PGCs were isolated and sequenced, along with associated somatic cells. The authors dealt with problem of very small number of migrating mouse PGCs by pooling cells from embryos (after ascertaining age matching using somite counting). 'Leading' and 'lagging' populations were separated by anterior and posterior embryo halves and the well-established Oct4-deltaPE-eGFP reporter mouse line was used.Weaknesses:The work seems to have been carefully done, but I do not feel the manuscript is very accessible, and I do not consider it well written. The novel findings are not easy to find. The addition of at least one figure to show the locations of putative signaling etc. would be welcome.

Thank you for the excellent suggestion. Fig. 6 has been added to highlight the main novel findings of this work and integrate them among contributions of earlier studies to provide a more complete view of signaling pathways and cell behaviors governing PGC migration.

(1) The initial discussion of CellRank analysis (under 'Transcriptomic shifts over developmental time...' heading) is somewhat confusing - e.g. If CellRank's 'pseudotime analysis' produces a result that seems surprising (some E9.5 cells remain in a terminal state with other E9.5 cells) and 'realtime analysis' produces something that makes more sense, is there any point including the pseudotime analysis (since you have cells from known timepoints)? Perhaps the 'batch effects' possible explanation (in Discussion) should be introduced here. Do we learn anything novel from this CellRank analysis? The 'genetic drivers' identified seem to be genes already known to be key to cell transitions during this period of development.

Thank you for this important observation. We have clarified the text in this section and added “This discrepancy may reflect differences in differentiation potential of some E9.5 PGCs that end in a terminal state among anterior E9.5 PGCs, but could also result from technical batch effects generated during library preparation. These possible interpretations are further discussed in the Discussion section.” to the pertinent results section and added additional relevant thoughts on the implications of this finding in Discussion paragraphs 4 and 7. We feel that it is important to include both results to the reader, as it is challenging to differentiate between heterogeneous developmental and migratory potential among E9.5 anterior PGCs and differential influence of batch effects across sequencing libraries with the data available.

(2) In Discussion - with respect to Y-chromosome correlation, it is not clear why this analysis would be done at E10.5, when E11.5 data is available (because some testis-specific effect might be more apparent at the later stage).

Since we had identified autocrine Nodal signaling primarily in anterior late migratory PGCs at E10.5 and knew that Nodal signaling was involved in sex specification of testicular germ cells into prospermatogonia by E12.5, we wanted to determine whether the Nodal signaling in late migratory PGCs at E10.5 was likely to be a sex-specific effect or was common to PGCs in both sexes. This was assessed in supplementary figure 4 and determined unlikely to be related to sex specification of PGCs as Nodal signaling was not strongly correlated with Y-chromosome transcripts in migratory PGCs. Assessing the relationship between Nodal signaling and Y-chromsome transcription at E11.5, when migration is complete, would be unlikely to help us further understand the dynamics of Nodal signaling during late PGC migration.

(3) Figure 2A - it seems surprising that there are two clusters of E9.5 anterior cells

Thank you for the interesting observation! One possibility is that the two states represent differential developmental competence as is suggested by the presence of one E9.5 anterior cluster along the differentiation trajectory in Fig 2A and one not within this differentiation trajectory. Another is that technical aspects of generating these sequencing libraries affected some cells more than others, resulting in clustering of highly affected and less affected cells, which would also be consistent with some E9.5 anterior cells lying within the differentiation trajectory and some not. Since it is challenging to differentiate between these possibilities with the data available, we have intentionally avoided overstating interpretations of this result in the manuscript text. We have included discussion of the potential implications of the transcriptional divergence you identify in Discussion paragraphs 4 and 7.

(4) Figure 5F - there does seem to be more LEFTY1/2 staining in the anterior region, but also more germ cells as highlighted by GFP

This is true; based on our selected anatomic landmarks for “anterior” and “posterior” as indicated in Methods, the “anterior” compartment typically contains more PGCs. Thus, we have included violin plots with all data points shown of signal intensities of both LEFTY1/2 and pSMAD2/3 in Fig. 5G and 5I so that the reader can evaluate the entire distribution of PGC signal intensities for each embryo.

**Reviewer #3 (Public review):**
Summary:The migration of primordial germ cells (PGCs) to the developing gonad is a poorly understood, yet essential step in reproductive development. Here, the authors examine whether there are differences in leading and lagging migratory PGCs using single-cell RNA sequencing of mouse embryos. Cleverly, the authors dissected embryonic trunks along the anterior-to-posterior axis prior to scRNAseq in order to distinguish leading and lagging migratory PGCs. After batch corrections, their analyses revealed several known and novel differences in gene expression within and around leading and lagging PGCs, intercellular signaling networks, as well as number of genes upregulated upon gonad colonization. The authors then compared their datasets with publicly available human datasets to identify common biological themes. Altogether, this rigorous study reveals several differences between leading and lagging migratory PGCs, hints at signatures for different fates among the population of migratory PGCs, and provides new potential markers for post-migratory PGCs in both humans and mice. While many of the interesting hypotheses that arise from this work are not extensively tested, these data provide a rich platform for future investigations.Strengths:The authors have successfully navigated significant technical challenges to obtain a substantial number of mouse migratory primordial germ cells for robust transcriptomic analysis. Here the authors were able to collect quality data on ~13,000 PGCs and ~7,800 surrounding somatic cells, which is ten times more PGCs than previous studies.The decision to physically separate leading and lagging primordial germ cells was clever and well-validated based on expected anterior-to-posterior transcriptional signatures.Within the PGCs and surrounding tissues, the authors found many gene expression dynamics they would expect to see both along the PGC migratory path as well as across developmental time, increasing confidence in the new differentially expressed genes they found.The comparison of their mouse-based migratory PGC datasets with existing human migratory PGC datasets is appreciated.The quality control, ambient RNA contamination elimination, batch correction, cell identification and analysis of scRNAseq data were thorough and well-done such that the new hypotheses and markers found through this study are dependable.The subsetting of cells in their trajectory analysis is appreciated, further strengthening their cell terminal state predictions.Weaknesses:Although it is useful to compare their mouse-based dataset with human datasets, the authors used two different analysis pipelines for each dataset. While this may have been due to the small number of cells in the human dataset as mentioned, it does make it difficult to compare them.

Direct comparisons between findings in human and mouse focused on CellChat cell-cell communication prediction results, which were conducted in an identical fashion using the same analysis methods for both datasets.

There were few validation experiments within this study. For one such experiment, whether there is a difference in pSMAD2/3 along the AP axis is unclear and not quantified as was nicely done for Lefty1/2.

Additional validation of the pSMAD2/3 signal intensity along the AP axis was performed and is now included in Fig. 5.